# Structural Causal Circuits: Probabilistic Circuits Climbing All Rungs of Pearl's Ladder of Causation

**Florian Peter Busch**                                    *florian_peter.busch@tu-darmstadt.de*
*Department of Computer Science*
*Technical University of Darmstadt*

**Moritz Willig**                                          *moritz.willig@cs.tu-darmstadt.de*
*Department of Computer Science*
*Technical University of Darmstadt*

**Matej Zečević**                                          *matej.zecevic@tu-darmstadt.de*
*Department of Computer Science*
*Technical University of Darmstadt*

**Kristian Kersting**                                      *kersting@cs.tu-darmstadt.de*
*Department of Computer Science*
*Technical University of Darmstadt*

**Devendra Singh Dhami**                                   *d.s.dhami@tue.nl*
*Department of Mathematics and Computer Science*
*Eindhoven University of Technology*

**Reviewed on OpenReview:** *https://openreview.net/forum?id=25XyUTICdZ*

## Abstract

The complexity and vastness of our world can require large models with numerous variables. Unfortunately, coming up with a model that is both accurate and able to provide predictions in a reasonable amount of time can prove difficult. One possibility to help overcome such problems is sum-product networks (SPNs), probabilistic models with the ability to tractably perform inference in linear time. In this paper, we extend SPNs' capabilities to the field of causality and introduce the family of structural causal circuits (SCCs), a type of SPNs capable of answering causal questions. Starting from conventional SPNs, we "climb the ladder of causation" and show how SCCs can represent not only observational but also interventional and counterfactual problems. We demonstrate successful application in different settings, ranging from simple binary variables to physics-based simulations.

## 1 Introduction

Consider the following example, which is an adaptation of a well-known scenario. Person $U$ keeps a small plant in their office but forgets to water it before going on a business trip. If person $U$ now remembers the plant ($U = 1$), a message $M$ is sent ($M = 1$) to two colleagues $A$ and $B$. Both colleagues water the plant ($A = 1$, $B = 1$) if they get a message, in which case the plant remains healthy ($H = 1$). Figure 1 shows the corresponding causal graph. In this example, a strong correlation between the plant being healthy and a message being sent can be observed, but it is clear that the plant's health has no causal impact on the message. To reason about causes and effects or to answer counterfactual questions such as "Given that the plant is healthy, would it still be healthy had $A$ not watered it?", an understanding of causality and its implications on the underlying structural equations is necessary.

Table 1: **Comparison of Models.** Types of causal queries (observational, interventional, counterfactual), which models are powerful enough to answer them, and whether inference is tractable.

| | Obs. $p(H\|A)$ | Interv. $p(H\| do(\neg A))$ | Counterf. $p(H_{do(\neg A)}\|A)$ | Tractability Full Evidence | Marginal |
|---|---|---|---|---|---|
| SPN (Poon & Domingos, 2011) | ✓ | ✗ | ✗ | ✓ | ✓ |
| iSPN (Zečević et al., 2021) | ✓ | ✓ | ✗ | ✓ | ✓ |
| Causal NF (Javaloy et al., 2024) | ✓ | ✓ | ✓ | ✓ | ✗ |
| NCM (Xia et al., 2023) | ✓ | ✓ | ✓ | ✗ | ✗ |
| cf-SPN (ours) | ✓ | ✓ | ✓ | ✓ | ✓ |

Causality can be seen as the science centered around the study of causes and effects (Pearl, 2009; Bareinboim & Pearl, 2016), which distinguishes between purely correlational observations and directed causal relations. Here, Pearl introduced the "ladder of causation" (Pearl & Mackenzie, 2018), which consists of an observational rung (correlations), an interventional rung (general causes and effects), and a counterfactual rung (hypothetical statements based on real-world evidence). Starting from the observational rung, each further step on the causal ladder describes a more difficult problem that requires more information to solve. Starting from the second rung, models can differentiate between the plant's health *being correlated with* 'sending a message' and the *directed causal impact* of the message on the plant.

Figure 1: **Watering Example.** Causal graph for the plant watering problem. $U$: owner remembers plant, $M$: owner sends message, $A,B$: colleague waters plant, $H$: plant is healthy. (Best viewed in color.)

An example of probabilistic models that can reason causally is Causal Bayesian networks (CBNs; Pearl (1995)). CBNs combine the advantages of Bayesian networks, i.e., decomposing the joint probability distribution into (interpretable) conditional distributions, with the field of causality and can thus reach the interventional, second rung of the causal ladder.

A major downside of CBNs is inference being intractable (Cooper, 1990). While approaches exist that circumvent this problem using approximations (Murphy et al., 2013), it would be desirable to obtain causal models that can perform *exact inference in tractable time.* To achieve this, sum-product networks (SPNs) (Poon & Domingos, 2011) pose a promising alternative as they specifically allow for exact tractable inference.

Unfortunately, conventional SPNs only operate on the observational rung of the ladder of causation. With the goal of utilizing the tractable inference property of SPNs in the field of causality, we take existing work of *interventional sum-product networks* (Zečević et al., 2021) leading us to 'climb one rung of the causal ladder' and introduce the novel class of counterfactual sum-product networks (cf-SPNs) to 'climb the final rung'. Overall, we propose the idea of *Structural Causal Circuits* (SCCs), a model family of SPNs capable of computing causal queries of any rung of Pearl's causal hierarchy. While there are other types of causal models, such as neural causal models (NCMs; Xia et al. (2023)) and causal normalizing flows (CNFs; Javaloy et al. (2024)), these either are intractable or lack the ability for tractable *marginal* inference (c.f. Table 1). The contributions of this paper are as follows:

1. We introduce cf-SPNs, tractable models capable of calculating probabilities for counterfactual queries by using a neural net (NN) to determine the SPN parameters for specific counterfactual worlds.

2. We introduce the model family of SCCs consisting of SPNs, iSPNs, and cf-SPNS, which consider the whole ladder of causation, enabling SPNs to answer questions on the entire Pearl's causal hierarchy.

3. We successfully apply SCCs on various problems, including discrete and continuous domains.

First, we go over the required background and related work in Section 2. Afterward, we introduce Structural Causal Circuits in Section 3. In Section 4, we start by experimentally comparing iSPNs and cf-SPNs and then demonstrate the capabilities of cf-SPNs in several experiments with varying problem difficulty.

## 2 Background and Related Work

Here, we explain the required background on causal models, the causal hierarchy, and SPNs, and give an overview of related work. We denote random variables by upper-case letters $V$, sets of random variables in boldface $\mathbf{V}$, values of single and sets of random variables as $v$ and $\mathbf{v}$ respectively, and probabilities of them as $P(v)$ or $P(\mathbf{v})$.

### 2.1 Causal Models

In order to introduce the necessary background on causality, we follow Pearl's formalism of causal models and say that

**Definition 1.** *A structural causal model (SCM) is a tuple $\mathcal{M} \coloneqq \langle \mathbf{V}, \mathbf{U}, \mathbf{F}, P_{\mathbf{U}} \rangle$ over a set of variables $\mathbf{X} = \{X_1, \ldots, X_K\}$ taking values in $\boldsymbol{\mathcal{X}} = \prod_{k \in \{1 \ldots K\}} \mathcal{X}_k$ subject to a strict partial order $<_{\mathbf{X}}$, where*

- $\mathbf{V} = \{X_1, \ldots, X_N\} \subseteq \mathbf{X}, N \leq K$ *is the set of endogenous variables,*

- $\mathbf{U} = \mathbf{X} \setminus \mathbf{V} = \{X_{N+1}, \ldots, X_K\}$ *is the set of exogenous variables,*

- $\mathbf{F} = \{f_1, \ldots, f_N\}$ *is the set of deterministic structural equations, i.e. $V_i \coloneqq f_i(\mathbf{X}')$ for $V_i \in \mathbf{V}$ and $\mathbf{X}' \subseteq \{X_j \in \mathbf{X} | X_j <_{\mathbf{X}} V_i\}$,*

- $P_{\mathbf{U}}$ *is the probability distribution over the exogenous variables $\mathbf{U}$.*

The relationships between the variables as described by $\mathbf{F}$ induce the directed graph $G(\mathcal{M})$, which by definition is acyclic due to $<_{\mathbf{X}}$. The exogenous variables $\mathbf{U}$ are usually unobserved. We say that an SCM $\mathcal{M}$ entails the probability distribution $P_{\mathbf{V}}^{\mathcal{M}}$ over the set of endogenous variables $\mathbf{V}$.

Interventions in causal models change how a variable value is determined, ignoring what was previously defined in the set of functions $\mathbf{F}$.

**Definition 2.** *Consider an SCM $\mathcal{M} \coloneqq \langle \mathbf{V}, \mathbf{U}, \mathbf{F}, P_{\mathbf{U}} \rangle$ and a variable $V_i \in \mathbf{V}$. Applying an intervention $do(V_i = v_i) \in \mathcal{I}$ on $\mathcal{M}$ replaces the structural equation $f_i$ with $\tilde{f}_i \coloneqq v_i$ and results in the intervened SCM $\mathcal{M}_{do(V_i = v_i)} \coloneqq \langle \mathbf{V}, \mathbf{U}, \tilde{\mathbf{F}}, P_{\mathbf{U}} \rangle$ where $\tilde{\mathbf{F}} = (\mathbf{F} \setminus \{f_i\}) \cup \{\tilde{f}_i \coloneqq v_i\}$.*

While naturally extendable to the multi-intervention case, we restrict our setting to a set of "allowed" interventions $\mathcal{I}$ for practical purposes, namely the set of all perfect interventions $\mathcal{I} = \{do(V_i = v_i) | V_i \in \mathbf{V} \wedge v_i \in \boldsymbol{\mathcal{X}}_i\}_{i \in \{1 \ldots N\}}$, i.e., interventions which set a variable to a constant value (Bongers et al., 2021). Every intervention induces a new mutilated graph $G(\mathcal{M}_{do(V_i = v_i)})$ to which we will refer to as $\tilde{G}$ for notational brevity. Every intervened causal model $\mathcal{M}_{do(V_i = v_i)}$ entails a new probability distribution $P_{\mathbf{V}}^{\mathcal{M}_{do(V_i = v_i)}}$.

One frequent assumption when using SCMs is the invariance of cause-effect relations (also known as invariance to the origin of the mechanism). In particular, a special type of invariance for interventions called "autonomy"[1] states that interventions should be local, i.e., with $\mathbf{PA}(V_i)$ denoting the set of direct parents of $V_i$ according to $G$ and for all $j \neq i$, it holds that

$$P^{\mathcal{M}}(V_j | \mathbf{PA}(V_j)) = P^{\mathcal{M}_{do(V_i = v_i)}}(V_j | \mathbf{PA}(V_j)). \tag{1}$$

That is, the conditional distributions of unintervened $V_j$ remain unchanged. This then allows for the truncated factorization of the SCM (Pearl, 2009)

$$P(\mathbf{V}) = \prod_{j \neq i} P(V_j | \mathbf{PA}(V_j)), \tag{2}$$

---

[1]For a short discussion on the meaning of "autonomy", see Appendix B.

suggesting independence between the intervened variable $V_i$ and its previous parents.

To extend our notion of SCMs to counterfactuals, we use the terminology of a "world" to describe a specific configuration of the entire set of endogenous variables.

**Definition 3.** *Consider an SCM $\mathcal{M} := \langle \mathbf{V}, \mathbf{U}, \mathbf{F}, P_{\mathbf{U}} \rangle$, an original world $\mathbf{V}' = \mathbf{v}'$, and an intervention $do(V_i = v_i) \in \mathcal{I}$. Due to the (counterfactual) intervention, we have $\tilde{\mathbf{F}} = (\mathbf{F} \setminus \{f_i\}) \cup \{\tilde{f}_i := v_i\}$. The distribution over the exogenous variables $P_{\mathbf{U}}$ is inferred to reproduce the original world $\mathbf{v}'$: $P_{\mathbf{U}}^{\mathbf{V}'=\mathbf{v}'} = P_{\mathbf{U}}(\mathbf{U}|\mathbf{V}' = \mathbf{v}')$. We call $\mathcal{M}_{do(V_i=v_i)}^{\mathbf{V}'=\mathbf{v}'} := \langle \mathbf{V}, \mathbf{U}, \tilde{\mathbf{F}}, P_{\mathbf{U}}^{\mathbf{V}'=\mathbf{v}'} \rangle$ the counterfactual SCM.*

In SCMs, the entire randomness responsible for sample variability is captured by $P_{\mathbf{U}}$ since all functions computing $\mathbf{V}$ are deterministic. In other words, each sample $\mathbf{u}$ entails a specific setting of variables $\mathbf{v}$. Thus, given the original world $\mathbf{v}'$, it is possible to infer information about $\mathbf{u}'$.[2] This is the meaning of the change from $P_{\mathbf{U}}$ of the original SCM to $P_{\mathbf{U}}^{\mathbf{V}'=\mathbf{v}'}$ of the counterfactual one: inferring the probability distribution of the exogenous variables given $\mathbf{v}'$. Applying an intervention now keeps all variables not influenced by the intervention fixed so that any counterfactual world $\mathbf{v}^*$ only differs from $\mathbf{v}'$ by the new value of the intervened variable itself, as well as all variables (and their descendants) incorporating the intervened variable via their structural equations. The resulting SCM $\mathcal{M}_{do(V_i=v_i)}^{\mathbf{V}'=\mathbf{v}'}$ represents the world counterfactual to $\mathbf{v}'$, had $V_i$ taken the value $v_i$ instead and entails the probability distribution $P_{\mathbf{V}}^{\mathcal{M}_{do(V_i=v_i)}^{\mathbf{V}'=\mathbf{v}'}}$.

Another perspective on counterfactuals is provided by "twin-networks" where each world is visualized as a separate causal graph with separate endogenous variables (Balke & Pearl, 1994). The exogenous variables are shared across both graphs. Counterfactuals can now be described by intervening on an endogenous variable in the counterfactual world.

## 2.2 The Causal Hierarchy

The set of functions $\mathbf{F}$ applied to a configuration of exogenous variables $\mathbf{u}$ yields a setting of endogenous variables $\mathbf{v}$, we write $\mathbf{F}(\mathbf{u}) = \mathbf{v}$. We say that for $\mathbf{Y} \subseteq \mathbf{V}$ with respective values $\mathbf{y} \subseteq \mathbf{v}$ and any $\mathbf{u}$, it holds that if $\mathbf{F}(\mathbf{u}) = \mathbf{v}$, then $\mathbf{Y}(\mathbf{u}) = \mathbf{y}$. In other words, under the set of exogenous variables $\mathbf{u}$, the variables $\mathbf{Y}$ take the values $\mathbf{y}$. Moreover, the probability $P_{\mathbf{U}}(\mathbf{Y} = \mathbf{y})$ represents the sum of probabilities over all endogenous variables $\mathbf{U}$ such that $\mathbf{Y} = \mathbf{y}$ follows from $\mathbf{U}$. This notation makes it possible to distinguish different causal queries according to their rung on the ladder of causation. To that end, Bareinboim et al. (2022) define the following symbolic languages:

**Definition 4** (Bareinboim et al. (2022))**.** *Let variables $\mathbf{V}$ be given and $\mathbf{W}, \mathbf{X}, \mathbf{Y}, \mathbf{Z} \subseteq \mathbf{V}$. Each language $\mathcal{L}_i, i = 1, 2, 3$, consists of (Boolean combinations of) inequalities between polynomials over terms $P(\alpha)$, where $P(\alpha)$ is an $\mathcal{L}_i$ term, defined as follows:*

- *$\mathcal{L}_1$ terms are those of the form $P(\mathbf{Y} = \mathbf{y})$, encoding the probability that $\mathbf{Y}$ take on values $\mathbf{y}$;*

- *$\mathcal{L}_2$ terms additionally include probabilities of interventions, $P(\mathbf{Y} = \mathbf{y}| do(\mathbf{X} = \mathbf{x}))$, giving the probability that variables $\mathbf{Y}$ would take on values $\mathbf{y}$, were $\mathbf{X}$ to have values $\mathbf{x}$;*

- *$\mathcal{L}_3$ terms encode probabilities over counterfactuals, $P_{\mathbf{U}^*}(\mathbf{Y}_{do(\mathbf{X}=\mathbf{x})} = \mathbf{y}, \ldots, \mathbf{Z}_{do(\mathbf{W}=\mathbf{w})} = \mathbf{z})$, where $\mathbf{U}^* = \{\mathbf{u}|\mathbf{Y}_{do(\mathbf{X}=\mathbf{x})}(\mathbf{u}) = \mathbf{y}, \ldots, \mathbf{Z}_{do(\mathbf{W}=\mathbf{w})}(\mathbf{u}) = \mathbf{z}\} \subset \mathbf{U}$, symbolizing the probability of variables $\mathbf{Y}, \ldots, \mathbf{Z}$ taking values $\mathbf{y}, \ldots, \mathbf{z}$, were $\mathbf{X}, \ldots, \mathbf{W}$ to have values $\mathbf{x}, \ldots, \mathbf{w}$, under values $\mathbf{u}$ consistent for all variables.*

For a model to be of a certain causal rung, it must be able to compute queries according to the desired symbolic language $\mathcal{L}_i$ (compare Definition 6). For notational brevity, we will write counterfactuals for a single counterfactual world $P_{\mathbf{U}^*}(\mathbf{Y}_{do(\mathbf{X}=\mathbf{x})} = \mathbf{y}, \mathbf{Z} = \mathbf{z})$, where $\mathbf{U}^* = \{\mathbf{u}|\mathbf{Y}_{do(\mathbf{X}=\mathbf{x})}(\mathbf{u}) = \mathbf{y}, \mathbf{Z}(\mathbf{u}) = \mathbf{z}\} \subset \mathbf{U}$ as $P(\mathbf{Y}_{do(\mathbf{X}=\mathbf{x})} = \mathbf{y}|\mathbf{Z} = \mathbf{z})$, highlighting the fact that the probability for the values $\mathbf{y}$ in the counterfactual world depends on the original world variables specified by $\mathbf{z}$.

---

[2]A step established as "abduction" in Pearl (2009).

### 2.3 Sum-Product Networks (SPNs)

An SPN (see an example in Figure 2, left) is a probabilistic graphical model consisting of a directed acyclic graph (DAG) and a set of weights. Each leaf represents a probability distribution over a variable, and multiple leaves can correspond to the same variable but contain different probability distributions. The inner nodes are either sum or product nodes. In a product node, the child probability distributions are multiplied and in a sum node, a weighted sum over the children is calculated.

The following definition considers binary variables as a means of illustrating the concepts for computing probabilities with SPNs. An extension to continuous variables is, without loss of generality, made possible by having continuous distributions in the leaf nodes of the SPN; for further reference, consider París et al. (2020). In our experimental section, we also include experiments on continuous variables. Formally, we can describe an SPN $\mathcal{S} = (G, \mathbf{w})$ by a DAG[3] $G = (V, E)$ and the non-negative weights $\mathbf{w}$. Sum and product nodes are given by $\mathsf{S}(\boldsymbol{\lambda}) = \sum_{\mathsf{C} \in \mathrm{ch}(\mathsf{S})} \mathbf{w}_{\mathsf{S},\mathsf{C}} \mathsf{C}(\boldsymbol{\lambda})$ and $\mathsf{P}(\boldsymbol{\lambda}) = \prod_{\mathsf{C} \in \mathrm{ch}(\mathsf{P})} \mathsf{C}(\boldsymbol{\lambda})$ , where $\boldsymbol{\lambda}$ is an indicator variable (IV). The SPN output is the value at the root node $\mathcal{S}(\boldsymbol{\lambda}) = \mathcal{S}(\mathbf{x})$ and probabilities can be computed by marginalization $P(\mathbf{x}) = \mathcal{S}(\mathbf{x})/\sum_{\mathbf{x}' \in \mathcal{X}} \mathcal{S}(\mathbf{x}')$.

The scope of a node is the set of variables that appear in the node or either of its children, i.e., for a node $\mathsf{N}$, the scope $\mathbf{sc}$ is defined by

$$\mathbf{sc}(\mathsf{N}) = \begin{cases} \{X\} & \text{if } \mathsf{N} \text{ is IV } (\boldsymbol{\lambda}_{X=x}) \\ \bigcup_{\mathsf{C} \in \mathrm{ch}(\mathsf{N})} \mathbf{sc}(\mathsf{C}) & \text{otherwise.} \end{cases} \tag{3}$$

*Completeness* is satisfied if all children of the same sum node have the same scope (Equation 4) and *decomposability* holds if no variable appears in multiple scopes of all children of the same product node (Equation 5) (Poon & Domingos, 2011):

$$\forall \mathsf{S} \in \mathcal{S} : (\forall \mathsf{C}_1, \mathsf{C}_2 \in \mathrm{ch}(\mathsf{S}) : \mathbf{sc}(\mathsf{C}_1) = \mathbf{sc}(\mathsf{C}_2)), \tag{4}$$

$$\forall \mathsf{P} \in \mathcal{S} : (\forall \mathsf{C}_1, \mathsf{C}_2 \in \mathrm{ch}(\mathsf{S}) : \mathsf{C}_1 \neq \mathsf{C}_2 \implies \mathbf{sc}(\mathsf{C}_1) \cap \mathbf{sc}(\mathsf{C}_2) = \emptyset). \tag{5}$$

In the scope of this paper, our primary emphasis revolves around enhancing the capability of sum-product networks (SPNs) to answer causal questions. Nevertheless, it is worth considering a broader perspective by exploring the encompassing definition of probabilistic circuits, as defined in Peharz et al. (2020a).

**Definition 5** (Peharz et al. (2020a)). *Given a set of random variables* $\mathbf{X}$*, a probabilistic circuit (PC)* $\mathcal{P}$ *is a tuple* $(\mathcal{G}, \psi)$*, where* $\mathcal{G}$*, denoted as computational graph, is an acyclic directed graph* $(V, E)$*, and* $\psi : V \mapsto 2^{\mathbf{X}}$*, denoted as scope function, is a function assigning some scope (i.e. a sub-set of* $\mathbf{X}$*) to each node in* $V$*. For internal nodes of* $\mathcal{G}$*, i.e. any node* $\mathsf{N} \in V$ *which has children, the scope function satisfies* $\psi(\mathsf{N}) = \cup_{\mathsf{N}' \in \mathbf{ch}(N)} \psi(\mathsf{N}')$*, where* $\mathbf{ch}(\mathsf{N})$ *denotes the set of children of* $\mathsf{N}$*. A leaf* $\mathsf{L}$ *of* $\mathcal{G}$ *computes a probability density over its scope* $\psi(\mathsf{L})$*. All internal nodes of* $\mathcal{G}$ *are either sum nodes (*$\mathsf{S}$*) or product nodes (*$\mathsf{P}$*). A sum node* $\mathsf{S}$ *computes a convex combination of its children, i.e.* $\mathsf{S} = \sum_{\mathsf{N} \in \mathbf{ch}(\mathsf{S})} w_{\mathsf{S},\mathsf{N}} \mathsf{N}$*, where* $\sum_{\mathsf{N} \in \mathbf{ch}\mathsf{S}} w_{\mathsf{S},\mathsf{N}} = 1$*, and* $\forall \mathsf{N} \in \mathbf{ch}(\mathsf{S}) : w_{\mathsf{S},\mathsf{N}} \geq 0$*. A product node* $\mathsf{P}$ *computes a product of its children, i.e.* $\mathsf{P} = \prod_{\mathsf{N} \in \mathbf{ch}(\mathsf{P})} \mathsf{N}$*.*

By this definition, an SPN is a PC. In contrast to arbitrary PCs, completeness and decomposability are required to hold for all SPNs.

### 2.4 Related Work

Related work exists on other models which compute counterfactual probabilities (Xia et al., 2023; Von Kügelgen et al., 2023; Bläser et al., 2025). As illustrated by further related work, arithmetic circuits (Darwiche, 2003) in general are a precursor to SPNs and have also been considered for a preliminary approach to causal inference (Darwiche, 2021). While Causal Bayesian networks are powerful and can be transformed into

---

[3]Not to be confused with a causal graph, which is also a DAG but not what is referred to here. While SPNs are usually depicted as trees, other DAG structures are also possible, e.g., by some nodes sharing the same parent nodes in the upper layers, see, for example, RAT-SPNs (Peharz et al., 2020b).

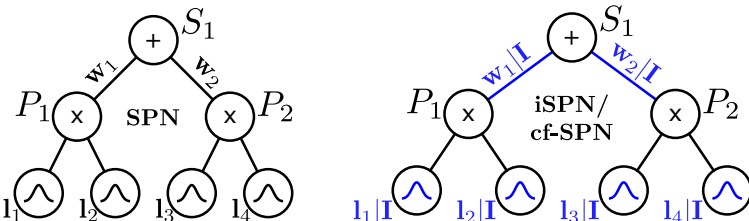

Figure 2: **SPN Examples. Left:** Observational SPN with fixed parameters. **Right:** In the iSPN and in the cf-SPN, the weights of the sum nodes **w** and the leaf distributions **l** are set according to the interventional information **I**, i.e., the intervention or counterfactual world (highlighted in blue). (Best viewed in color.)

SPNs and back (Zhao et al., 2015), this transformation from SPNs generally leads to degenerate[4] Bayesian Networks incapable of subsequent causal inference (Papantonis & Belle, 2020). However, a model class extension, as used in this paper, poses a viable candidate for overcoming the problems of using SPNs for causal inference. More recently, Papantonisa & Belle (2023) introduced an algorithm to transform SPNs into Bayesian networks, which simplifies the calculation of interventional queries. Studying the complexity of counterfactuals, Han et al. (2022) highlighted that calculating counterfactuals using circuits is not any more complex than interventional or observational questions. Huber et al. (2023) also consider counterfactuals in circuits by investigating partial identifiability. A compositional perspective on inference in probabilistic circuits allows inferring interventional distributions (Wang & Kwiatkowska, 2023). Recently, normalizing flows have also been used for causality (Javaloy et al., 2024) but lack the capability to perform marginal inference or only consider small problems (Chen et al., 2024).

Recent work has used SPNs for generating counterfactuals for prediction tasks (Němeček et al., 2025), but they do consider counterfactual predictions from an explainable artificial intelligence perspective and do not consider a wider range of counterfactuals for probabilistic problems. Shao et al. (2020) introduced *conditional SPNs* where an NN is used to set the SPN parameters depending on some set of conditional variables that are input in the NN. Building on top of this work, Zečević et al. (2021) introduced *interventional SPNs* (iSPNs) which use the same idea of a combined NN and SPN architecture to allow for interventional queries. Causal circuits have been shown to help with scaling (Busch et al., 2024). In this paper, we take iSPNs and extend them further to cf-SPNs (counterfactuals SPNs) by providing the necessary causal information into an NN setting the SPN parameters (see Figure 2 on an example of how parameters are set in SPNs, iSPNs, cf-SPNs). In combination, we end up with a family of causal SPNs that span the entirety of Pearl's ladder of causation: Structural Causal Circuits (SCCs).

## 3 Structural Causal Circuits

Known for their tractable inference properties, SPNs have mainly been employed to answer queries on observational data. In the following, we outline a path that sees SPNs as "climbing up the ladder" of causation, thereby forming a family of structural causal circuits capable of answering causal questions. To do so, we must first define when a model can be considered to answer observational, interventional, or counterfactual queries. We make use of the definition of symbolic languages by Bareinboim et al. (2022) that link the particular syntactic expression of a query to a respective rung on the causal ladder. We require models to be able to truthfully answer those queries to be considered part of a specific rung.

**Definition 6.** *Structural Causal Circuits (SCCs) are a family of probabilistic circuits $\{SCC_1, SCC_2, SCC_3\}$ able to answer causal queries defined by the symbolic languages $\mathcal{L}_i$ of Definition 4. For a model $m : \mathbf{V} \to \mathbb{R}$ containing a PC over variables $\mathbf{V}$, we say that $P^m$ is the probability distribution entailed by $m$ and we define*

- $m \in SCC_1$ *iff* $\forall P(\mathbf{V} = \mathbf{v}) \in \mathcal{L}_1(\mathcal{M}).P^{m(\mathbf{V})} = P_{\mathbf{V}}^{\mathcal{M}}$

- $m \in SCC_2$ *iff* $\forall P(\mathbf{V} = \mathbf{v} | do(V_i = v_i)) \in \mathcal{L}_2(\mathcal{M}).P^{m(\mathbf{V}, do(V_i = v_i))} = P_{\mathbf{V}}^{\mathcal{M}_{do(V_i = v_i)}}$

---

[4]A bipartite graph in which the actual variables of interest are not connected is called degenerate.

- $m \in SCC_3$ *iff* $\forall P(\mathbf{V}_{do(V_i=v_i)} = \mathbf{v}|\mathbf{V}' = \mathbf{v}') \in \mathcal{L}_3(\mathcal{M}).P^{m(\mathbf{V},do(V_i=v_i),\mathbf{V}'=\mathbf{v}')} = P_{\mathbf{V}}^{\mathcal{M}_{do(V_i=v_i)}^{\mathbf{V}'=\mathbf{v}'}}$

*where $\mathcal{M}$ is any SCM.*

In plain terms, a model $m$ is part of the respective class $SCC_1/SCC_2/SCC_3$ if it can represent the observational/interventional/counterfactual probability distribution of the particular rung. To this end, an $SCC_1$ model takes a setting of variables $\mathbf{V}$ as input. An $SCC_2$ model takes additional intervention information $do(V_i = v_i)$, and an $SCC_3$ model additionally gets provided with the original world setting. We write $\mathbf{V}' = \mathbf{v}'$ to indicate a single fully specified original world. We will show in Section 4.2 that models of lower rungs cannot answer causal queries of higher rungs correctly since they do not integrate the additionally required interventional or counterfactual information in their inference process. Most common counterfactual queries consider one original and one counterfactual world, which is why we opted for this definition due to ease of notation. The definition could be adapted by adding a variable number of worlds that are not required to be fully specified in order to support any type of counterfactual query as in Definition 4. The definition requires matching the probability distribution of "some" coherent SCM, since multiple SCMs can generate the same data. Therefore, it is unreasonable to expect any model to learn the "true" SCM for a specific problem.

Previous work on inference using tractable circuits shows that exact computation of interventional distributions is #P-hard (Wang & Kwiatkowska, 2023). While this is true for complex distributions, simple problems (distributions) can still be computed exactly and efficiently. Simple graph structures (for example, chains) or properties of the data distribution can be leveraged to compute queries more efficiently. Thus, we argue that practical applications rarely represent the worst-case scenario, but rather feature compressible distributions that facilitate the training of an SCC that, therefore, supports tractable computation of queries. Distributions are well compressible by an SCC if there are many (context-specific) independencies that the SCC can leverage. In this case, even small SCCs can learn complex problems well or even exactly (Martens & Medabalimi, 2014). Otherwise, increasingly larger SCCs can be employed to approximate more complex probability distributions arbitrarily well. The possible problems to be encountered do not lie so much in the realm of causal representability but are rather related to the engineering side of setting up and training SCC. Generally, our definition states that if a model is found to fulfill the above criteria, it is part of the respective family of SCCs. Another practical constraint concerns the approximate behavior of our models. We require the probability distributions of model $m$ and SCM $\mathcal{M}$ to be equal. In practice, however, training is performed with a finite amount of data, making *perfect* matching of the distribution impossible. We, therefore, relax the equality constraint and say that our models should "truthfully approximate" the probability distribution up to some small $\epsilon$. Consider also that in some scenarios, domain experts could potentially set up models with the exact weights, yielding a model that matches the causal distribution perfectly.

### 3.1 Observational Sum-Product Networks

The first type of SCCs does not require any causal knowledge. Here, we can consider conventional SPNs and make the following definition.

**Definition 7.** *We refer to any SPN that does not incorporate interventional or counterfactual data to answer a probabilistic query as an* observational SPN*.*

In other words, an SPN models an observational probability distribution and is, therefore, to be placed on the first rung of the ladder of causation.

Reconsider the plant watering example of Figure 1. For the sake of a simple example sufficient for conveying the key concepts, it is assumed that all relations are deterministic (this is just for simplicity and not a necessary assumption). Say that $U$ is an exogenous, unobserved variable that is true with some probability, and all other variables are endogenous. We have $\mathbf{V} = \{M, A, B, H\}$ and $\mathbf{U} = \{U\}$. For this watering problem, one is able to deduce $U$ from any other variable since all assignments are deterministic and bijective. Note that this is not always true, for example, if the relation between $U$ and $M$ was not deterministic, and person $U$ might remember the plant but still not write the message. In this deterministic case, however, *observing* the problem boils down to only two possible configurations: either person $U$ did not remember, which leads

to all other variables being false as well and the plant ending up in an unhealthy state, or person $U$ did remember, all variables are true, and the plant ends up healthy.

Let us say an SPN is learned using data for the observable variables of the watering problem. Several leaves, each representing an observable variable, would be created and combined into alternating product and then sum nodes until the final root sum node of the SPN is reached. In our example, the SPN would have $\mathbf{sc}(\mathsf{N}) = \mathbf{V} = \{M, A, B, H\}$ as its scope and be able to compute any probability of the joint distribution of those variables, e. g. $P(W, A, B, H)$ or $P(\neg A)$.

Generally speaking, we have

**Proposition 1.** *All observational SPNs are in $SCC_1$.*

*Proof.* Observational SPNs are complete and decomposable, and therefore valid (Poon & Domingos, 2011). By definition, valid SPNs approximate normalized distributions (up to numerical error). Specifically, valid SPNs are able to approximate the observational distribution $P^{\mathcal{M}}$ of some SCM $\mathcal{M}$. All queries $P(\mathbf{V} = \mathbf{v}) \in \mathcal{L}_1$ are terms over $P^{\mathcal{M}}$ and thus can be answered by an observational SPN. □

Now, what if $A$ takes care of the plant anyway, independent of whether its owner wrote a message or not? Or, knowing that the plant ended up healthy, would it still be healthy had $B$ not watered it? In these easy examples, the consequences are obvious, but they can not be computed by a purely observational SPN, as it lacks the required vocabulary and information to ask and answer such questions. The following section (3.2) goes into detail on how *interventional* queries differ from *observational* ones and which changes to an SPN can be made to allow for answering such queries.

### 3.2 Interventional Sum-Product Networks

Let us address the first of the two concluding questions from the previous section, namely "What if $A$ has always taken care of the plant anyway?". Considering the causal graph, this intervention represents removing all edges going towards the variable to be intervened upon, indicating that $A$ is no longer determined by its parents. In this example, we would see a scenario where the plant is always healthy, independent of $U$. Generally, interventions can greatly change probabilities compared to the observational setting. The observational SPN lacks a mechanism to answer interventional queries as it only models the observational joint probability distribution. There is simply no way to ask an observational SPN the question of $P(H|do(A))$, as no kind of inference results in the correct answer.[5] Having been trained only on correlational data, an observational SPN lacks the necessary causal information about the effects of interventions.

An SPN capable of answering interventional queries must, therefore, have a source of information about the second rung of the causal ladder and an appropriate mechanism to utilize that knowledge and make answering interventional queries possible. To this end, the following architecture for an *interventional sum-product network* (iSPN) was proposed by Zečević et al. (2021).

**Definition 8** (Zečević et al. (2021)). *An* interventional sum-product network (iSPN) *is the joint model* $m(\mathbf{G}, \mathbf{D}) = g(\mathbf{D}; \boldsymbol{\psi} = f(\mathbf{G}; \boldsymbol{\theta}))$, *where* $g(\cdot)$ *is a SPN,* $f(\cdot)$ *a non-parametric function approximator and* $\boldsymbol{\psi} = f(\mathbf{G})$ *are shared parameters.*

The (mutilated) causal graph $\mathbf{G} \in \{0, 1\}^{N \times N}$ is given as input into the neural network $f$ so that the model $m$ can estimate the density of the given data matrix $\{\mathbf{V}_k\}_k^K = \mathbf{D} \in \mathbb{R}^{K \times N}$ by learning the parameters $\boldsymbol{\theta}$, where $K$ is the number of data points and $N$ is the number of variables.

For example, given the question $P(H|do(A))$, the NN would get the intervention $do(A)$ as an input and return the parameters $\boldsymbol{\psi}$ (leaf distributions and sum node weights) of the SPN. Now, the SPN can be queried just like an observational SPN, but each query is conditioned on the intervention. Asking for $P(H)$ on the intervened upon SPN is the answer to the query $P(H|do(A))$. In other words, each intervention creates an observational SPN of the world where that intervention takes place.

---

[5]Except when an interventional question can be transformed into an observational one, which, of course, can be calculated by an observational SPN. The exact conditions for when this can be inferred follow from the do-calculus (Pearl, 2009).

From the original paper on iSPNs (Zečević et al., 2021), we have for an intervention $do(\mathbf{V}_j = \mathbf{v}_j)$, data matrix $\{\mathbf{V}_k\}_k^K = \mathbf{D} \in \mathbb{R}^{K \times N}$ and causal graph $G$:

**Proposition 2** (Zečević et al. (2021)). *Assuming autonomy and invariance, an iSPN $m(\mathbf{G}, \mathbf{D})$ is able to identify any interventional ($\mathcal{L}_2$) distribution $P^{\mathcal{M}}(\mathbf{V}_i = \mathbf{v}_i | do(\mathbf{V}_j = \mathbf{v}_j))$, permitted by a SCM $\mathcal{M}$ through interventions, with knowledge of the mutilated causal graph $\hat{G}$ and data $\mathbf{D}$ generated from the intervened SCMs by modelling the conditional distribution $P^{\mathcal{M}_{do(\mathbf{V}_j = \mathbf{v}_j)}}(\mathbf{V}_i = \mathbf{v}_i | \mathbf{V}_j = \mathbf{v}_j)$. Any iSPN is part of $SCC_2$.*

Therefore, an iSPN is able to correctly calculate that $P(H | do(A)) = 1$.

### 3.3 Counterfactual Sum-Product Networks

Let us now consider the last rung of the causal ladder by imagining the question, "Given that we know that person $B$ watered the plant, would the plant still be healthy had person $B$ not watered the plant?". It is clear that the answer to this question is given neither by the observational rung 1 query $P(H | \neg B)$ nor by the interventional rung 2 query $P(H | do(\neg B))$, as both queries can not express the current and counterfactual state of $B$ simultaneously. The right way to express the question is via the counterfactual formulation $P(H_{do(\neg B)} | B)$. The query asks for the counterfactual value of $H$ under an intervention that sets $B$ to false, given that $B$ was true in the original world. To give the right answer, a model needs to infer the state of $A$ from the actual state of $B$, and incorporate the inferred knowledge about $A$ into the counterfactual world where $B$ is intervened. However, to an iSPN, which does not differentiate between actual and counterfactual values, $B$ can only be true or false (or unknown). Serving the intervention $do(\neg B)$ to an iSPN while simultaneously conditioning it $B = true$ results in a probability of 0, which is the correct answer, considering the interventional query $P(B | do(\neg B)) = 0$, i.e., "What is the probability of $B$ being true if $B$ is set to false?".

The proposed *counterfactual sum-product network* (cf-SPN) expands upon the iSPN and enables SPN to answer counterfactual queries. We use an asterisk (*) to indicate variables of the counterfactual world.

**Definition 9.** *A* counterfactual sum-product network (cf-SPN) *is the joint model $m(\mathbf{G}, \mathbf{D}) = g(\mathbf{D}^*; \boldsymbol{\psi} = f(\mathbf{D}', \mathbf{G}; \boldsymbol{\theta}))$, where $g(\cdot)$ is an SPN, $f(\cdot)$ a non-parametric function approximator, $\boldsymbol{\psi} = f(\mathbf{D}', \mathbf{G})$ are shared parameters of the SPN, $\mathbf{D}' \in \mathbb{R}^{K \times N}$ and $\mathbf{D}^* \in \mathbb{R}^{K \times N}$ are data matrices with observational and counterfactual values, respectively, and $\mathbf{D} = (\mathbf{D}', \mathbf{D}^*)$.*

$\mathbf{G} \in \{0,1\}^{N \times N}$ is the (mutilated) causal graph according to some intervention $do(\mathbf{V}_j^* = \mathbf{v}_j^*)$. We use counterfactual data to train the model, such that the data matrix $\mathbf{D} \in \mathbb{R}^{K \times 2N}$ contains pairs of observational $\mathbf{D}' = \{\mathbf{V}_k'\}_k^K \in \mathbb{R}^{K \times N}$ and counterfactual $\mathbf{D}^* = \{\mathbf{V}_k^*\}_k^K \in \mathbb{R}^{K \times N}$ variable settings. This definition assumes complete evidence of both the observational and the counterfactual world.

Generally, one can view cf-SPNs as part of a family of partially causal models (PCMs) (Zečević et al., 2023). In practice, one might be unable to provide full data on all possible counterfactual settings during cf-SPN training. In such cases, cf-SPNs might only be trained to answer particular subsets of counterfactual queries (e.g., all queries containing a single intervention). Still, we present limited generalization results of cf-SPN beyond their training distribution in Section 4.6.

As with iSPNs, a neural network, which is provided with interventional information $\mathbf{G}$, is used to determine the parameters of the SPN. However, cf-SPNs extend the NN input by concatenating the setting of variables for the original world $\mathbf{D}'$ to it. This extension adds the required inputs to provide the value of the original world as a separate input while simultaneously being able to reason over the counterfactual world at the SPN inputs, therefore lifting the cf-SPN above the interventional level.

**Proposition 3.** *Assuming autonomy and invariance, a cf-SPN $m(\mathbf{G}, \mathbf{D})$ is able to identify any counterfactual ($\mathcal{L}_3$) distribution $P^{\mathcal{M}}(\mathbf{V}_i^* = \mathbf{v}_i^* | \mathbf{V}' = \mathbf{v}', do(\mathbf{V}_j^* = \mathbf{v}_j^*))$, permitted by a SCM $\mathcal{M}$ through counterfactuals, with knowledge of the mutilated graph $G^*$, the original world variables $\mathbf{v}' \in \mathbf{D}'$ generated from the original SCM, and corresponding counterfactual data $\mathbf{v}^* \in \mathbf{D}^*$ by modelling the distribution $P^{\mathcal{M}_{do(V_i = v_i)}^{\mathbf{V}' = \mathbf{v}'}}(\mathbf{V}_i^* = \mathbf{v}_i^* | \mathbf{V}_j' = \mathbf{v}_j')$. Any cf-SPN is part of $SCC_3$.*

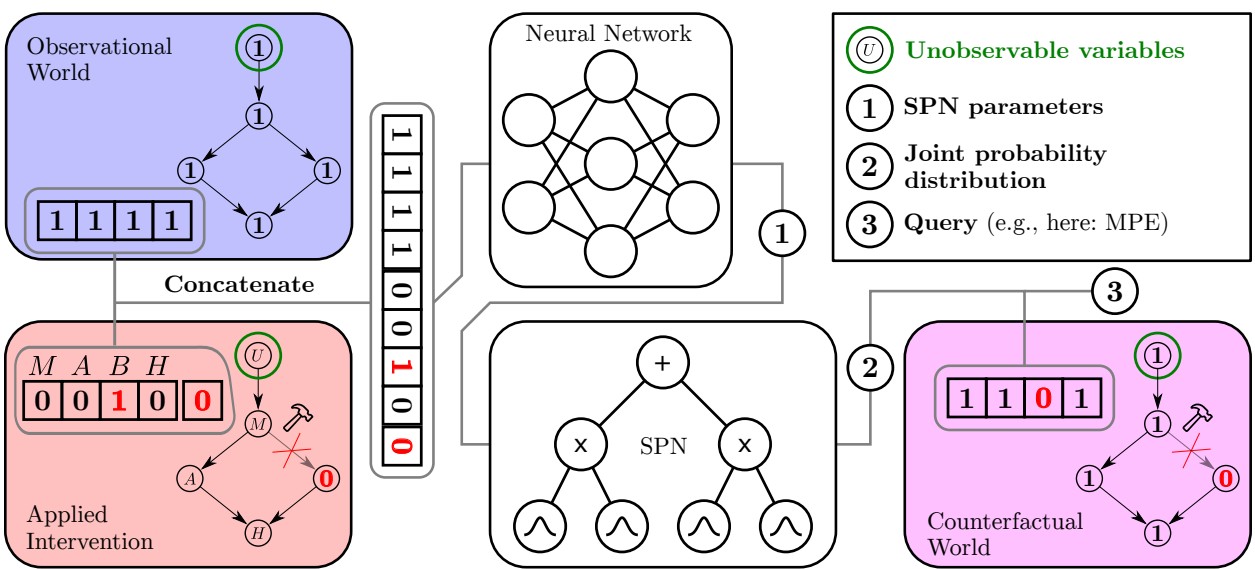

Figure 3: **Counterfactual SPN.** Computation of the counterfactual query $\arg\max_{\mathbf{X}} P(\mathbf{X}_{do(\neg B)}|M, A, B, H)$ using a cf-SPN. Information about the observational world (blue box) and the intervention (red box; indicating the intervened upon variables and their new value) is concatenated and given to a NN, which in turn computes the SPN parameters. The parameterized SPN can then be evaluated to answer counterfactual questions about a counterfactual world (violet box). (Best viewed in color.)

*Proof.* Let $\mathcal{M} \coloneqq \langle \mathbf{V}, \mathbf{U}, \mathbf{F}, P_{\mathbf{U}} \rangle$ be the observational SCM. From the *do*-calculus (Pearl, 2009), we know that $P^{\mathcal{M}}(\mathbf{V}_i^* = \mathbf{v}_i^* | \, do(\mathbf{V}_j^* = \mathbf{v}_j^*)) = P^{\mathcal{M}_{do(\mathbf{V}_j' = \mathbf{v}_j')}}(\mathbf{V}_i^* = \mathbf{v}_i^* | \mathbf{V}_j^* = \mathbf{v}_j^*)$. The counterfactual SCM $\mathcal{M}_{do(V_i = v_i)}^{\mathbf{V}' = \mathbf{v}'} \coloneqq \langle \mathbf{V}, \mathbf{U}, \tilde{\mathbf{F}}, P_{\mathbf{U}}^{\mathbf{V}' = \mathbf{v}'} \rangle$ is equal to the interventional SCM $\mathcal{M}_{do(V_i = v_i)} \coloneqq \langle \mathbf{V}, \mathbf{U}, \tilde{\mathbf{F}}, P_{\mathbf{U}} \rangle$ if $P_{\mathbf{U}}^{\mathbf{V}' = \mathbf{v}'} = P_{\mathbf{U}}$. For the specific sample $\mathbf{V}' = \mathbf{v}'$, we have $P_{\mathbf{U}}(\mathbf{U}|\mathbf{V}' = \mathbf{v}') = P_{\mathbf{U}}^{\mathbf{V}' = \mathbf{v}'}$. It remains to be shown that an SPN can learn the joint probability distribution $P(\mathbf{V}^*)$, which follows from Poon & Domingos (2011). $\square$

The full flow of computation for a cf-SPN is illustrated in Figure 3. All original setting variables in the aforementioned example are set to true, and the variable $B$ is intervened with a value of zero. Both vectors are concatenated and given to the NN, which outputs the parameters for the SPN $\boldsymbol{\psi}$. The resulting SPN estimates the distribution of the counterfactual world, such that all queries to the SPN are of a counterfactual nature. In our example, the desired probability for $H$ would be 1, indicating a 100% probability that the plant would still be healthy. This is the correct prediction as $A$ would still have watered the plant, even if $B$ would have been prevented from doing so.

Note that counterfactual data is required for training a cf-SPN. This is generally unobtainable, as it is generally impossible to directly "measure" values from the counterfactual outcome. This situation can be addressed in multiple ways. First, our model shows that SPNs are capable of answering counterfactual queries if the correct model is given, illustrating the potential of the proposed approach for domain-specific, expert-engineered models. Second, as shown in other works on counterfactual models (e.g., Xia et al. (2023)), it is sometimes possible to calculate counterfactuals when only being provided with information from the lower rungs of the causal ladder. Furthermore, several methods exist that can be used to approximate counterfactual outcomes.

### 3.4 Learning Structural Causal Circuits

We assume the architecture of the SPN to be given and use RAT-SPNs (Peharz et al., 2020b) in our experiments to set the architecture. How to strategically learn the structure of SCCs is left for future work. SCCs are trained similarly to conventional SPNs. The learning objective is to maximize the log-likelihood of the data given a dataset $\mathbf{D} \in \mathbb{R}^{K \times N}$ with $K$ instances and $N$ features. For the observational SPN,

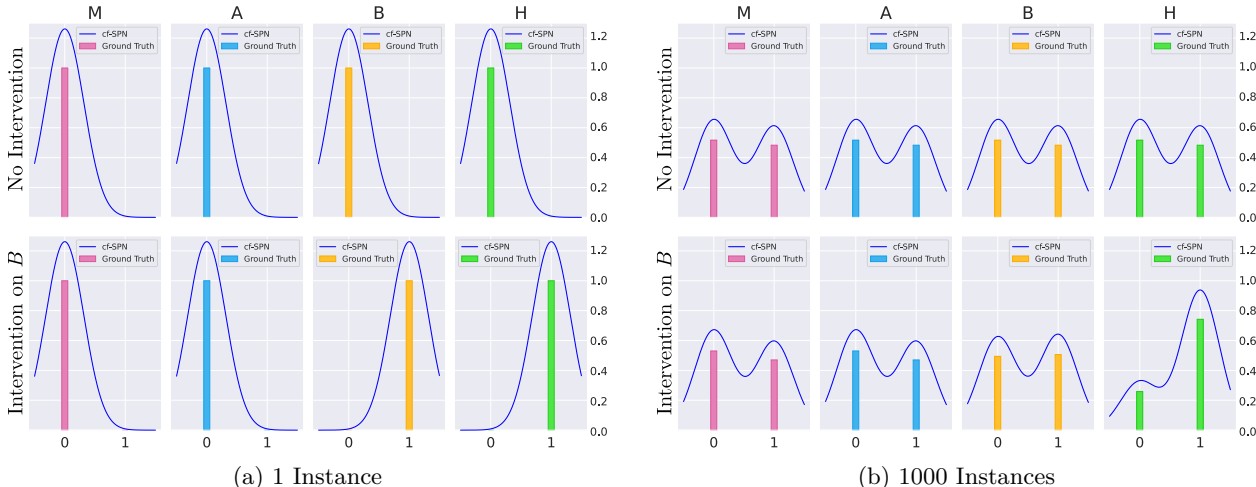

Figure 4: **cf-SPN Watering Experiment Results.** The cf-SPN predictions (blue lines) match the ground truth (bars) without any noticeable error. With the additional information about the original world, counterfactual statements can be made reliably even about single instances. (Best viewed in color.)

we can simply backpropagate through the SPN nodes to learn the correct parameters (see also Poon & Domingos (2011)). The interventional as well as the counterfactual SPN can be learned end-to-end using backpropagation, where the leaf distributions and sum node weights are not learned as fixed parameters but instead set through the preceding neural network. Therefore, we do not learn the parameters $\boldsymbol{\psi}$ directly, but backpropagate into the NN and optimize its parameters $\boldsymbol{\theta}$.

The acquisition of counterfactual data can be challenging in practice. Multiple approaches, such as propensity matching (Rosenbaum & Rubin, 1983), quasi-experiments (Rutter, 2007), or "difference-in-differences" (Liu et al., 2021) can be used to approximate counterfactual data. We go into further detail on this topic and provide a brief discussion in Appendix C.

## 4 Experiments

There already exists plenty of work demonstrating the application of SPNs for the observational setting. We kindly refer the reader to the survey of Sánchez-Cauce et al. (2021) for a more comprehensive overview. In our evaluation, we will therefore focus specifically on interventional and, in particular, counterfactual queries. We will briefly revisit the watering example from the introduction and continue with an experiment that showcases the inability of iSPNs to correctly model counterfactual distributions, while the cf-SPN is able to do so. Finally, we present two physics-related experiments, showcasing the use of cf-SPN for particle movements and simulated galaxy collisions. In all experiments, we use a RAT-SPN (Peharz et al., 2020b) to create the graph of the SPN and assume Gaussian distributions in the leaf nodes. We make our code publicly available at `https://github.com/olfub/SCC`.

### 4.1 Watering Example

In this experiment, we revisit the watering example from the introduction to explain the behavior of the cf-SPN in detail (c.f. Figures 1 and 3). The root variable $U$ is true 50% of the time, and all other variables $M, A, B$, and $H$ follow from it deterministically. The input to the NN of our cf-SPN consists of the intervention information (i.e., both the intervention target variable and the intervention value to be set) and a single configuration of the original world variables. The output of the NN is then used to parameterize the SPN. The model is trained using counterfactual data by providing pairs of instances where the original world, the counterfactual world, and the intervention that distinguishes the counterfactual from the original world

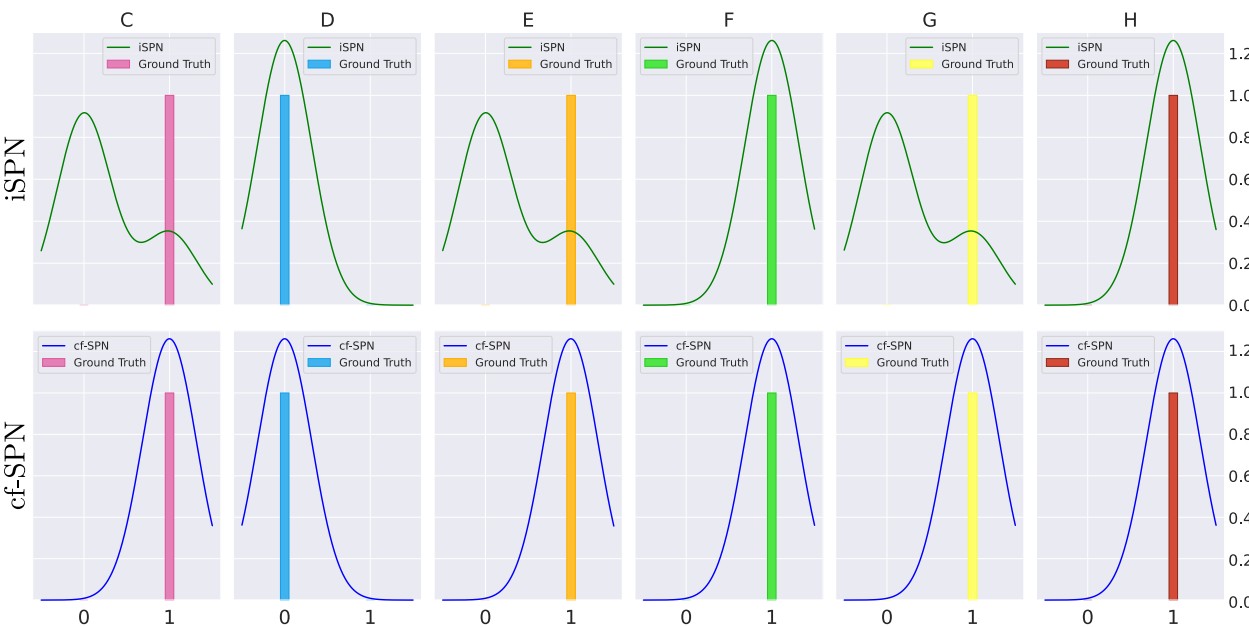

Figure 5: **Counterfactual Predictions of iSPN and cf-SPN.** The original world setting is $\{A = 1, B = 1, C = 1, D = 0, E = 1, F = 1, G = 1, H = 1\}$. $D$ is intervened upon ($do(D = 0)$). The iSPN (green lines) does not consider the original world setting, therefore averaging predictions and failing to match the ground truth. The cf-SPN (blue lines) incorporates the original world setting and can, therefore, predict the correct counterfactual outcome of the counterfactual query. (Best viewed in color.)

are known. As the SPN and the NN are differentiable, both can be optimized with an end-to-end training procedure. (Compare for Appendix D, describing the training setups for this and all following experiments.)

Figure 4 shows results for an intervention on variable $B$. (Refer to Appendix E.3 for all results.) Because we use Gaussian leaves within the SPN, the model predicts continuous probability densities. We plot the densities in the range of $[−0.5, 1.5]$ (lines) while displaying the ground truth as discrete values (bars). For Figure 4a, all variables in the original world are set to false and –without an intervention– the counterfactual world is identical to the original world. When specifying an intervention on $B$, which sets it to 1, the cf-SPN infers that $H = 1$ is the correct counterfactual outcome. An intervention on $B$ indicates that person $B$ is set to water the plant, independent of getting the message $M$ or $A$ watering the plant. The predictions of the cf-SPN for $M$ or $A$ are therefore kept correctly unaltered.

Figure 4b follows the same type of setup but averaged over 1000 different original world instances with a 50% chance of an intervention $do(B = 1)$. Without interventions, $U$ (and thus all other variables) would have been set to 0 and 1 half of the time. However, considering the average distribution over 1000 samples, $H$ is set to 1 more often. This is because the intervention on $B$ introduces a new possibility for $H$ being 1, namely the scenario where $U$ (and therefore $H$) would have been set to zero, but the counterfactual intervention on $B$ sets $B$, and therefore $H$, to 1.

## 4.2 Answering Counterfactual Queries with cf-SPN

Consider the causal graph depicted in Figure 7.[6] The exogenous variables $A$ and $B$ are unobserved and random with $P(A) = 0.7$ and $P(B) = 0.4$ from which all other variables follow deterministically. We use an iSPN and a cf-SPN to calculate the resulting counterfactual probabilities for each variable, given the intervention $do(D = 0)$ and the specific original world setting that follows from $\{A = 1, B = 1\}$ using the equations in Figure 7. Results are shown in Figure 5. The iSPN is only able to predict the general interventional distribution, while the cf-SPN matches the ground truth accurately. As discussed before,

---

[6]The causal graph was taken from https://plato.stanford.edu/entries/counterfactuals/.

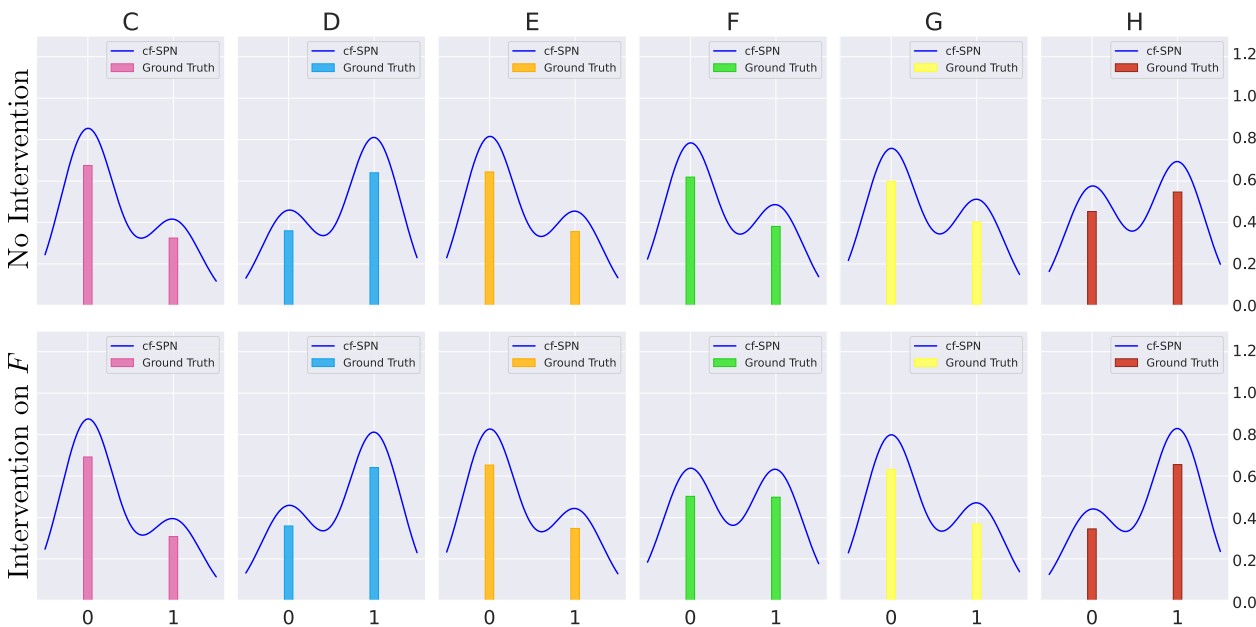

Figure 6: **cf-SPN Experiment with Noise Results.** "Noisy" setup with a 10% probability of switching the resulting value of a function to its opposite value. The cf-SPN prediction (blue lines) matches the ground truth (bars) without any significant error, showing the success of the model to learn the distributions correctly. All graphs were created by sampling 1000 instances. (Best viewed in color.)

this is not surprising as the iSPN does not incorporate the sample-specific information of the original world and only predicts the general interventional distribution for $do(D = 0)$. On the other hand, the cf-SPN considers the additional 'original world' input, which allows the model to make predictions about the specific counterfactual outcome. Generally, the cf-SPN is able to match the ground truth well, while the iSPN is forced to resort to predicting averaged probabilities.

To demonstrate that cf-SPN can also deal with non-deterministic environments, we can also consider the slightly more challenging setting where the same logical operators as before are applied, but modified to give opposite results 10% of the time. We show the results for an exemplary intervention on the variable $F$ in Figure 6. (Results for interventions on all other variables are presented in Appendix E.2.) The ground truth probability masses are given by the bar charts and are computed as the average over 1000 samples. Similarly, the averaged model predictions are computed over the same 1000 samples and plotted as blue lines. We can see how a uniform, random intervention on $F$ increases the probability of $H$ being true while $C$, $D$, $E$, and $G$ remain unchanged. This

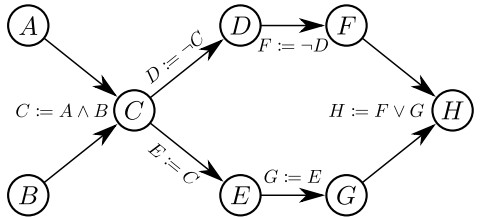

Figure 7: **Exemplary Causal Graph.** $A$ and $B$ are treated as unobserved, exogenous variables. In one variation, all assignments are deterministic. In another variation, the underlying functions are the same, but each variable has a chance of 10% to take on the opposite value.

noisy dataset is learned without noticeable error by the cf-SPN. But shouldn't there be some uncertainty due to the noise? No! As all noise –e.g., due to external factors– is already encoded in the original world setting, the cf-SPN should be able to make an accurate prediction and answer the counterfactual queries perfectly and without error, which it does successfully.

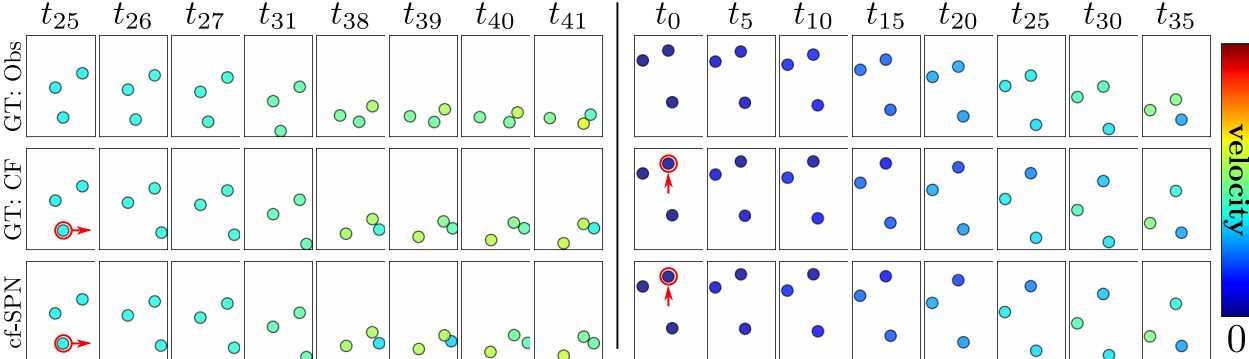

Figure 8: **cf-SPN Particle Collision Experiment. GT: Obs**: simulation without intervention, **GT: CF**: simulation with intervention, **cf-SPN**: cf-SPN prediction for the intervention. Left: move a particle to the right on $t_{26}$. Right: set the velocity of a particle slightly upwards ($t_0$). The model prediction matches the simulation very well, with only some visible differences when it comes to collisions. (Best viewed in color.)

## 4.3 Particle Collision

This more challenging, continuous problem is based on a simulation for particles moving and colliding in a box[7]. Here, the size of the box, the number and radius of particles, and the forces between them (collision and external accelerations) are fixed. Three particles are used, which can be described by four values each: vertical position, horizontal position, vertical velocity, and horizontal velocity. Using this simulation, we can both run a simulation without intervention and run the same simulation but, at some time step, set a value to some specific other value, i.e., apply an intervention. Considering two such runs, the second one is counterfactual to the first one, for example, "How would the particles have behaved, had one specific particle been moved to another spot after some time steps?".

In order to highlight the expressiveness and flexibility of cf-SPNs, we slightly alter the experimental setup compared to the previous ones. Instead of only inputting the original world at the time of an intervention and training an SPN to predict the resulting counterfactual state, which only differs from the original one by means of the counterfactual intervention, we additionally learn the simulation step of the counterfactual state. This additional task, which involves predicting the particles' movement with respect to their velocity, changes what the NN must learn and enables the cf-SPN to run full simulations by itself.

In the experiments, an acceleration force (gravity) is dragging the particles to the bottom, and all particles start with a velocity of 0 at time step 0. Particles can collide with each other and with the walls of the box. In Figure 8, two simulations are shown where the overall velocity of the particles is indicated by their color, ranging from dark blue (0 velocity) over green and yellow to red (increasingly faster).[8] In the first row (GT: Obs), the simulation is shown, which runs for 50 time steps without interventions, i.e., a rung 1 simulation. The second row (GT: CF) also shows data from the simulation, but at a certain time step, an intervention occurs, i.e., this is the counterfactual simulation. Finally, the third row (cf-SPN) shows the same as the row above (GT: CF), but now, instead of using the simulation, the cf-SPN model is used to predict the most likely next positions and velocities. The third row (cf-SPN) should ideally be identical to the respective ground truth shown above in the second row.

For the first example (Figure 8, left), the particles start at some position further up and then slowly but steadily accelerate downwards for 25 time steps. At this point in time ($t_{25}$), the bottom particle is moved to the right (indicated as an intervention with a red circle and arrow), where it can be found one time step later at $t_{26}$. The particle behavior is learned by the cf-SPN, with only some small deviation when it comes to collisions. In the second example (right half of Figure 8), the intervention takes place immediately, this time not changing a position but the velocity in the vertical direction (indicated by a red arrow "pushing" that particle upwards). While this velocity is relatively quickly negated by the downwards acceleration, a

---

[7]https://github.com/ineporozhnii/particles_in_a_box
[8]See https://github.com/olfub/SCC/blob/main/PaperGifs.md for the gifs.

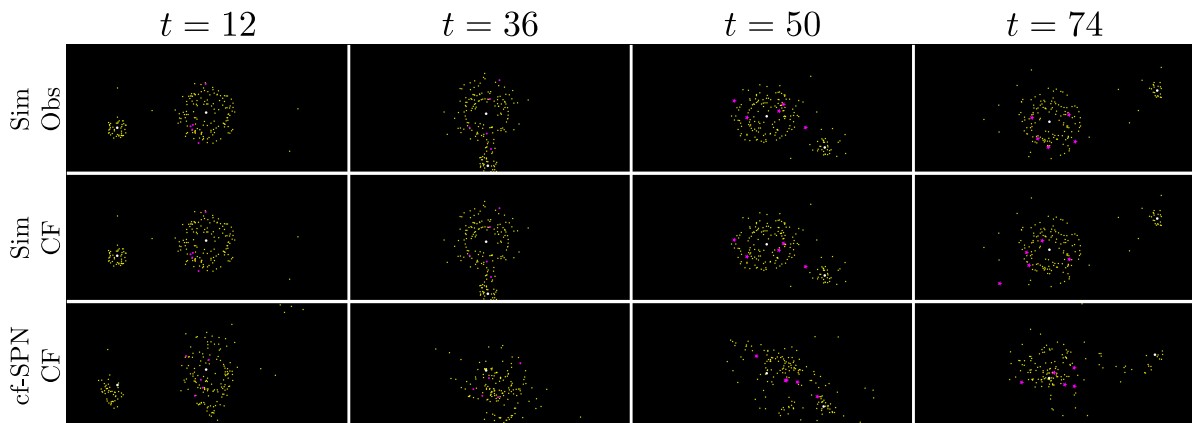

Figure 9: **cf-SPN Galaxies Experiment.** Interventions on the positions of some particles (magenta) between time step 49 and time step 50 move these particles down, such that the unfolding of the system is altered in the following. The cf-SPN is able to reflect the changes of the intervention and roughly simulate the consequent unfolding of the simulation. (Best viewed in color.)

clear difference between this intervened case (GT: CF) and the non-intervened behavior (GT: Obs) can be seen for later time steps. In this example without any collisions, the cf-SPN predictions show no visible differences to the simulation.

### 4.4 Galaxies

Upscaling the problem from before, we now consider the collision of galaxies with stars (particles) orbiting around black holes. Using a publicly available simulation[9], the semi-implicit Euler is used to calculate a ground truth simulation where a galaxy with a smaller mass comes into contact with a galaxy with a heavier black hole. As a result, some particles from the smaller galaxy join the larger one while others are thrown out of orbit.

We adopt a similar approach to the particle collision experiment. Black holes and particles are encoded using Euclidean coordinates and velocities among these coordinates. Since individual particles are only influenced by the black holes (no inter-particle gravity), it suffices to input both black holes along with one particular particle to predict its position and velocity at the next time step. Multiple particles are predicted by repeating the single-particle simulation, using the same black hole values to end up with a full simulation step. In a perfect model, the black hole prediction would be independent of the particle information. Since the black hole position is predicted with every particle prediction, we average all predictions to calculate the black holes' positions and velocities for the next time step. Our experiment focuses on a single trajectory of these two black holes. Even though this does simplify the problem, learning the gravitational forces simulated in the model, combined with the escalating propagation of even minor errors in earlier time steps, still poses a major challenge. As part of a hyperparameter optimization (see Appendix D.1 for more information), we found that the best results were achieved using an NN without any hidden layers.

From top to bottom, the three rows in Figure 9 show the simulation without intervention, the simulation where an intervention occurs in time step 49, and the prediction of the interventional scenario by the cf-SPN.[10] The intervention moves 5 particles (stars; colored magenta) downwards while leaving their velocities unaltered. Over all time steps, the model predictions deviate slightly from the simulation, leading to a steady accumulation of smaller errors over time. The interventions themselves, however, are roughly matched with the particles' position changing in the correct direction. Despite several challenges, the two galaxies and a significant portion of the particles are predicted to lie in the correct regions even after more than 70

---

[9]https://github.com/EnguerranVidal/GalaxyCollision

[10]See https://github.com/olfub/SCC/blob/main/PaperGifs.md for an animation of the simulation, and Appendix E.5 for additional time steps.

| | Method | 5/5 | 10/12 | 15/20 | 20/30 | 50/100 | 100/250 |
|---|---|---|---|---|---|---|---|
| ID | cf-SPN | 0.99 ± 0.01 | 0.99 ± 0.00 | 0.97 ± 0.01 | N/A | N/A | N/A |
| | CNF | 0.98 ± 0.01 | 0.98 ± 0.01 | 0.97 ± 0.02 | N/A | N/A | N/A |
| OOD | cf-SPN | 0.97 ± 0.02 | 0.98 ± 0.01 | 0.98 ± 0.00 | 0.99 ± 0.00 | 0.99 ± 0.00 | 0.98 ± 0.00 |
| | CNF | 0.99 ± 0.01 | 0.99 ± 0.00 | 0.99 ± 0.00 | 0.99 ± 0.00 | 0.99 ± 0.00 | 0.99 ± 0.00 |

Table 2: **Element-Wise Accuracy Comparing cf-SPNs and CNFs.** Different combinations of nodes/edges are evaluated, ranging from 5 to 100 nodes. Some counterfactual queries have been part of the training data (ID), others have not (OOD). Both models predict the values of single variables correctly almost every time. CNFs perform slightly better on OOD data.

| | Method | 5/5 | 10/12 | 15/20 | 20/30 | 50/100 | 100/250 |
|---|---|---|---|---|---|---|---|
| ID | cf-SPN | 0.27 ± 0.20 | 0.34 ± 0.20 | 0.86 ± 0.90 | N/A | N/A | N/A |
| | CNF | 6.84 ± 6.94 | 1.72 ± 0.92 | 1.28 ± 1.32 | N/A | N/A | N/A |
| OOD | cf-SPN | 2.56 ± 1.36 | 0.60 ± 0.35 | 0.60 ± 0.23 | 0.54 ± 0.13 | 0.91 ± 0.07 | 1.90 ± 0.28 |
| | CNF | 13.20 ± 7.40 | 4.38 ± 1.91 | 10.09 ± 6.84 | 4.77 ± 2.91 | 1.31 ± 0.10 | 1.19 ± 0.27 |

Table 3: **L2-Errors of Marginal Probabilities for cf-SPNs and CNFs (scaled by 100).** Different combinations of nodes/edges are evaluated, ranging from 5 to 100 nodes. Some counterfactual queries have been part of the training data (ID), others have not (OOD). This table shows the average L2-error for the marginal probability per variable times 100 (as most differences are quite small). ID samples tend to be predicted with lower errors, and the cf-SPN gives smaller errors throughout all but the largest setting.

time steps, which –considering the propagating errors– is a noteworthy achievement. We acknowledge the challenge of learning complex, larger problems and are looking at scaling causal SPNs in future work.

### 4.5 Model Comparison

To the best of our knowledge, SCCs are the only models capable of performing tractable marginal inference for causal queries (Bareinboim et al., 2022). There are not many other models that are suitable for comparison. In this section, we compare cf-SPNs with two other models for causal inference, namely causal Bayesian networks (CBN) (Pearl, 1995) and causal normalizing flows (CNF) (Javaloy et al., 2024). These models are close to cf-SPNs in scope; however, CBNs do not support tractable marginal inference, and CNFs can not perform marginal inference at all. In addition, both models require the causal graph as an input.

For these experiments, we generate random DAGs with binary variables and random conditional probability tables, given a specified number of variables (nodes) and edges. We sample counterfactual data from these graphs and then train a cf-SPN on this data, in the same manner as for the experiments described in previous sections. Additionally, a CBN is trained on the counterfactual data and the causal graph, and a CNF is trained on the observational data and the causal graph. Both models require the causal graph to be known, which cf-SPNs do not. We evaluate these models by randomly sampling 100 unique original worlds and intervention combinations for each experiment. The CBN is then used to calculate the marginal probabilities for each variable using the variable elimination algorithm.[11] We use the resulting CBN marginal probabilities as the gold standard prediction and measure the error of cf-SPN and CNF outputs against it. While doing so, the inference time of all models is measured and recorded. Note that CNFs do not support marginal queries and only return counterfactual samples. We can (and will) use these samples and interpret them as marginal probabilities by interpreting the counterfactual samples as the marginal probabilities. This is not entirely unreasonable, as we are dealing with binary variables and, therefore, a sample of 0/1 could reasonably represent a marginal probability of 0/1 for the respective variable. Of course, this approach is still not technically correct (these are not probabilities), but it serves as a proxy given the CNF's general inability to compute marginal probabilities. While this allows us to compute the error between the expected (CBN) marginal probability and the model (cf-SPN, CNF) predictions, we also employ another evaluation

---

[11]We only evaluate on 100 unique original worlds as otherwise this CBN evaluation takes too much time for the larger problems (see Table 4).

| Method | 5/5 | 10/12 | 15/20 | 20/30 | 50/100 | 100/250 |
|--------|-----|-------|-------|-------|--------|---------|
| cf-SPN | 1.05 $\pm 0.04$ | 2.10 $\pm 0.05$ | 3.07 $\pm 0.06$ | 3.80 $\pm 0.20$ | 9.92 $\pm 1.00$ | 18.47 $\pm 1.05$ |
| CBN | 1.00 $\pm 0.02$ | 2.10 $\pm 0.08$ | 3.55 $\pm 0.54$ | 22.47 $\pm 25.73$ | 17.68 $\pm 2.03$ | 1395.52 $\pm 2194.34$ |
| CNF | 0.33 $\pm 0.01$ | 0.52 $\pm 0.01$ | 0.70 $\pm 0.02$ | 0.81 $\pm 0.05$ | 1.90 $\pm 0.20$ | 3.35 $\pm 0.18$ |

Table 4: **Average Inference Runtimes for CBNs and cf-SPNs (seconds). CNFs are faster but do not compute marginal probabilities, but only single samples.** Shown are the inference times for computing all marginal probabilities in an evaluation, divided by the number of variables. While the runtime scales roughly linearly for cf-SPNs, the computation for CBNs is much less efficient because of its intractability. The runtime decreases from 20 to 50 nodes as we use slightly different evaluation strategies. More information can be found in Appendix D.2.

metric that fairly assesses the CNF prediction quality by considering the most probable explanation instead of marginal probabilities. To this end, we map all probabilities or samples larger than 0.5 to 1 and all others to 0 and record the resulting predictive accuracy. We conduct experiments with the following number of nodes/edges across 5 seeds each: 5/5, 10/12, 15/20, 20/30, 50/100, 100/250. Further details and explanations on the experimental setup and choice of evaluation are included in Appendix D.2.

Table 2 shows the element-wise accuracy of the most likely counterfactual values.[12] For each input sample, we check whether it also existed as such in the training data, which then determines whether it will be included as in-distribution (ID) or out-of-distribution (OOD). Both cf-SPNs and CNFs perform well overall, both when the query was part of the training data and when it was not. CNFs sometimes perform slightly better, but the differences are very small throughout. Note that CNFs are trained using the ground truth causal graphs. In practical scenarios, it is often unlikely that the true graph is discovered from the data perfectly. In Table 3, the average L2-errors (times 100) of the marginal probabilities are shown. Firstly, we can see how now out-of-distribution samples are predicted with larger error for both models (with one exception for 15/20). Additionally, we can now see how CNFs' outputs are less useful when interpreted as marginal probabilities. While doing so is inherently unfaithful, CNFs do not support the computation of marginal probabilities directly, which is why we had to use such a proxy. With this proxy, however, cf-SPNs perform much better, especially on out-of-distribution samples, where interpreting CNF samples as probabilities is inherently flawed, as can be seen from the small configurations in particular. We include two additional metrics in Appendix E.4.

In Table 4, the average inference times per variable are shown. CNFs are faster than both CBNs and cf-SPNs, but they do not compute marginal probabilities and, therefore, their fast inference times should not be compared directly to CBNs and cf-SPNs. Both these models are close to each other at first, but the tractability property of cf-SPNs allows them to remain much faster than CBNs for larger problems, scaling approximately linearly, while the inference time for CBNs explodes for larger problems.

## 4.6 Prediction on Unseen Inputs

We revisit the noise problem of Section 4.2 to further analyze our model's generalization capabilities. In our first experiment for testing on unseen inputs, we take the same training data as before but remove the $k$ least frequent variable settings, i.e., data points where the original world has these variable settings are not included.[13] We use a special evaluation test set that contains all possible inputs once, i.e., any original world paired with any single intervention, resulting in 832 inputs. Since we only consider counterfactuals, we can ask the model for the most probable explanation and compare its answer with the ground truth. This way, we judge our model's performance by measuring its accuracy. Results are shown in Table 5a. Even if 48 of the possible 64 combinations are not part of the training data, the model is still able to predict the majority of samples correctly, showcasing its potential to adapt to unseen instances.

---

[12]Only two seeds for 5/5 contain OOD samples, therefore, the mean and standard deviation are only computed using a few evaluations on these two seeds.

[13]Note that, due to binary variables, there are $2^6 = 64$ possible variable settings in this problem.

Table 5: **Prediction on Unseen Inputs. (a):** Out of 64 possible variable settings, $k$ are excluded from the training data. The accuracy is calculated based on how often the model predicts the most probable counterfactual outcome correctly. Even if most inputs are unseen when training, accuracy remains relatively high. **(b) and (c):** The accuracy shows the model predictions when a new number of interventions are considered. For training, the model only saw instances with 1 and 2 (b) or 1 and 3 interventions (c).

(a) Unseen Worlds

| $k$ | Accuracy |
|---|---|
| 0 | $1.00 \pm 0.00$ |
| 8 | $0.99 \pm 0.00$ |
| 16 | $0.99 \pm 0.01$ |
| 24 | $0.98 \pm 0.01$ |
| 32 | $0.95 \pm 0.02$ |
| 40 | $0.90 \pm 0.01$ |
| 48 | $0.83 \pm 0.03$ |
| 56 | $0.52 \pm 0.03$ |

(b) Multiple Interventions.
Trained on 1 and 2 Interventions.

| $m$ | Accuracy |
|---|---|
| 1 | $0.96 \pm 0.01$ |
| 2 | $0.92 \pm 0.01$ |
| 3 | $0.83 \pm 0.04$ |
| 4 | $0.72 \pm 0.06$ |
| 5 | $0.61 \pm 0.09$ |
| 6 | $0.52 \pm 0.14$ |

(c) Multiple Interventions.
Trained on 1 and 3 Interventions.

| $m$ | Accuracy |
|---|---|
| 1 | $0.89 \pm 0.03$ |
| 2 | $0.84 \pm 0.03$ |
| 3 | $0.86 \pm 0.03$ |
| 4 | $0.89 \pm 0.03$ |
| 5 | $0.91 \pm 0.03$ |
| 6 | $0.91 \pm 0.07$ |

Secondly, we take a look at how well the cf-SPN manages to go beyond single interventions. To this end, we first change the model input. Instead of a single one-hot encoder with one additional field for the intervention values, each variable that supports interventions now has two fields: one for indicating whether an intervention takes place, and another one containing the intervention value itself. Now, we train one model on data that contains interventions on 1 or 2 variables and another model on data containing interventions on 1 or 3 variables. We evaluate all possible inputs for $m \in [1, 2, 3, 4, 5, 6]$ counterfactual interventions (where 6 indicates an intervention on every variable) and record the accuracy in the same manner as for the previous experiment. From the results in Tables 5b and 5c, we can see how the model trained on up to 2 interventions generalizes with mixed success, while a model trained on 1 and 3 interventions does not drop in accuracy for the higher number of interventions ($m$). We conclude that the cf-SPN model can successfully generalize to new inputs that have never been seen as part of the training data.

## 5 Conclusion

In this paper, we introduced SCCs, a novel class of models that utilize the tractability of SPNs for causal settings, therefore "climbing" the ladder of causation. We demonstrated how iSPNs can learn interventional distributions but fail when it comes to counterfactual predictions. To this end, we introduced the novel class of cf-SPN and demonstrated throughout several experiments that this more powerful class of SPNs succeeds in learning counterfactual distributions, highlighting the expressiveness of cf-SPNs on various problems with tabular data and in settings of particle and galaxy collisions. Having shown the potential of SCCs across all rungs of the ladder of causation, we can look towards improving the practical applicability of SCCs even further. Our current models require data from the respective rung of the causal ladder to train them. While some causal information will always be necessary for models beyond the observational level, one could try to utilize different information, for example by including domain knowledge or counterfactual samples gained from human experts. Another challenge when applying these models is scalability and expressiveness for more difficult problems with a large number of variables.

**Limitations.** The presentations learned by SCCs do not represent a semantic structure similar to that of structural causal models, but rather present an efficient representation of the distributions' partition function (Martens & Medabalimi, 2014). While this might complicate validation of the learned mechanisms, our experiments show good coherence to the underlying ground truth data. As with all approaches that rely on learning mechanisms purely from data, adequate amounts of data with sufficient variability are required for the model to identify the underlying mechanisms and converge to the true distribution. For the individual case, corresponding testing and validations should be put in place to secure the safe operation of the model.

**Acknowledgments**

This work is supported by the Hessian Ministry of Higher Education, Research, Science and the Arts (HMWK; project "The Third Wave of AI"), the ICT-48 Network of AI Research Excellence Center "TAILOR" (EU Horizon 2020, GA No 952215) and by the Federal Ministry of Education and Research (BMBF; project "PlexPlain", FKZ 01IS19081). The authors acknowledge the support of the German Science Foundation (DFG) research grant "Tractable Neuro-Causal Models" (KE 1686/8-1). The Eindhoven University of Technology authors received support from their Department of Mathematics and Computer Science and the Eindhoven Artificial Intelligence Systems Institute.

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

## Appendix

In this appendix, we first elaborate on the tractability of SCCs (Appendix A) and the definition of "Autonomy" used in this paper (Appendix B). After that, we discuss the possibilities of acquiring counterfactual data for training a cf-SPN (Appendix C). We then elaborate on our experimental section of the main paper by first giving additional details on the experimental setup (Appendix D), then providing additional results (Appendix E), and, lastly, giving some technical details regarding the environment in which experiments were run (Appendix F).

## A  Tractability and Expressivity of SCCs

We motivated our work on causal SPNs by having tractable models for answering causal questions. Observational SPNs are tractable (Poon & Domingos, 2011), being able to answer queries in time linear in the size $d$ of the model, that is, $O(d)$. Considering iSPN and cf-SPN, the NN time depends on its architecture. For a simple NN, the runtime is mainly determined by the matrix-vector multiplications, so the complexity depends cubically on the size of the input $N$, resulting in $O(N^2)$. Any query can then be answered within $O(N^2d)$ which, if the intervention $G$ of the iSPN or the counterfactual world of the cf-SPN (described by the intervention and the original world, i.e., $G^*$, $\mathbf{v}' \in \mathbf{D}'$) remains unchanged, reduces to $O(d)$ since the NN pass does not need to be repeated for every query.

Overall, SPNs are tractable partly because they only use simple arithmetic operations for any inference. Product nodes (multiplication) are responsible for the SPN's ability to represent simple distributions very efficiently: if parts of the data are independent, their probabilities can simply be multiplied. In the most trivial case, an SPN could even only consist of a single product node and no sum nodes; this would represent a probability distribution where all variables are independent from each other. Since this will almost never be true in practice (and if it is, one would not need to employ an SPN), sum nodes are the other SPN component to increase the expressivity of the model. In other words, by splitting the data into different regions (sum nodes), the SPN can leverage independencies that only hold in such regions: context-specific independencies. By combining these two aspects, SPNs are capable of learning or approximating many distributions very well.

On the other hand, not every single distribution can be represented *perfectly* with an arbitrary SPN. Since underlying operations such as marginalization are inherently intractable, tractability can not always be achieved without compromise. However, if distributions are well compressible, i.e., they contain context-specific independencies that the SPN can leverage, then SPNs can learn such distributions with no or only a small margin of error, meaning that all distributions can be approximated up to an $\epsilon$ of error. For iSPN and cf-SPN in particular, the NN component is also a universal function approximator, ensuring that SCCs can learn any distribution of their respective rung.

## B  Interpretations of "Autonomy"

We are aware of two notions of autonomy. In Peters et al. (2017), the authors refer to Aldrich (1989), stating that in Aldrich's notion "autonomous relations are likely to be more stable than others. He equates Haavelmo's autonomous variables with what subsequently became known as exogenous variables. Autonomous variables are parameters fixed by external forces or treated as stochastically independent." In other words, this notion of autonomy describes the exogenous variables, i.e., the "noise variables" or the randomness of the model. Our notion of autonomy (also described in Peters et al. (2017)) describes the invariance with respect to interventions, i.e., that conditional distributions of unintervened variables remain unchanged from interventions on different variables.

## C  Acquisition of Counterfactual Data

The main benefit of cf-SPNs lies in their aforementioned tractability, allowing for fast inference of a variety of queries. However, counterfactual data is required for training such a model. Often, we are unable to

acquire "truly" counterfactual data as, by definition, only the factual outcome can be observed, while the counterfactual outcome stays hidden. While true counterfactuals can only be obtained in a very limited number of settings, several approaches exist to remedy this kind of situation. First of all, it is always possible to utilize preexisting knowledge of domain experts to construct an SPN. Similar to how BNs and other graphical models can be set up explicitly, we can also construct (i/cf-)SPNs with their corresponding conditioning mechanisms that truthfully represent the underlying causal relations.

However, there might arise situations where we are unable to access expert knowledge. Thus, our SPNs have to be trained from data. Several approaches exist and are in widespread use as of today to approximate the potential outcome of counterfactual events (Rubin, 2005). While randomized controlled trials (Chalmers et al., 1981) are acknowledged as the gold standard for estimating overall effects, they do not help us with individual-level causal effects. One idea to remedy this situation is the use of propensity score matching (Rosenbaum & Rubin, 1983). The key idea is to pair up similar participants/data points, assuming that their starting conditions (speak - latent variables $\mathbf{u}$) are also the same. Apart from analyzing actual but scarcely available twin studies (McCartney et al., 1990; McGue et al., 2010), this method can help to create 'statistical' twins that share the same initial properties. This allows us to observe multiple factual outcomes, where one can be seen as the counterfactual to the other. Research on this topic includes analysis methods such as quasi-experiments (Rutter, 2007) in general and the "difference-in-differences" (Liu et al., 2021) method in particular, and is still applied in today's modern systems (e.g., online advertising (Bottou et al., 2013)).

As long as we hold the means of acquiring such counterfactual data to a degree that allows for practical application by revealing causal relations and/or breaking spurious confounding, we are guaranteed to approximate the true underlying distribution up to numerical error and the expressive capacity of our model.

If counterfactual data is approximated with noise or errors, the cf-SPN will learn the distribution as given by the data. In other words, if the data is noisy, the cf-SPN predictions will be more uncertain, and if the data is faulty, the cf-SPN will assume this to be representative of the true distribution. The challenge of obtaining good counterfactual data is, therefore, separate from the learning task of the cf-SPN.

**Practical Limitations.** Most commonly, counterfactual queries are computed with respect to a single "factual" world. While $\mathcal{L}_3$ (Definition 4) permits multiple such factual worlds, we are restricting ourselves to the former scenario and consider data with fully specified original and counterfactual variables, where the counterfactual differs from the observation by means of a single intervention.

## D   Experimental Details

Additional details on the experimental setup are shown in Table 6. All experiments use the Adam optimizer (Kingma & Ba, 2014) with a learning rate of 0.001. We always use Gaussian distributions in the leaves for their ability to represent continuous distributions. When computing probabilities for binary variables, we normalize the densities for 0 and 1 to obtain the probabilities for 0 and 1. Gradient clipping is applied to Galaxies experiments, where gradients are clipped to a value of 0.5 for numerical stability. For the particles experiment (*), an intervention can be applied to any of the 12 variables. (Three particles defined via four attributes [position X, position Y, velocity X, velocity Y]). The positions can be set to some value within the box and the velocity has a minimum and maximum possible value as well. For the data generation process, an intervention is chosen randomly and uniformly within its domain. Several simulations are run where random timesteps are stored in the data. In a single simulation, most data points are discarded so that the datasets consist of a variety of different data points. This is balanced in such a way that sufficiently many interventions are included in the data. For the galaxies experiment (**), each prediction step only considers two black holes and a single star. Only the position of the stars can be intervened upon, moving them a fixed amount to the top/bottom/left/right. The information about the original world here describes the previous timestep, but since we only simulate a specific collision course, the space of possible inputs is not exhausted. All interventions take place at the same time in the simulation, and some are randomly selected to be part of the training data.

Table 6: **Experimental Setup Details.** The *Interventional* dataset was shown in Section E.1, the variations *CF without Noise* and *CF with Noise* in Section 4.2 and Appendix E.2, *Watering* in Section 4.1 and Appendix E.3, *Particles* in Section 4.3, and *Galaxies* in Section 4.4 and Appendix E.5. #V, #Train/test, and #E show the number of variables, the train/test set size, and the training epochs. $\mathcal{I}$ indicates the set of interventions, i.e., on how many variables can be intervened upon. **V** describes the input for the original world in the cf-SPN.

| Dataset Name | #V | #Train/Test | #E | $\mathcal{I}$ | **V** |
|---|---|---|---|---|---|
| Interventional | 6 | 560000/140000 | 50 | Any | - |
| CF without Noise | 6 | 560000/140000 | 50 | Any | Any |
| CF with Noise | 6 | 560000/140000 | 50 | Any | Any |
| Watering | 4 | 400000/100000 | 50 | Any | Any |
| Particles | 12 | 192000/48000 | 100 | Any* | Any* |
| Galaxies | 12 | 96000/24000 | 100 | 2** | Trajectory** |

### D.1 Galaxy Collision Details

For the galaxy collision experiment, we systematically explored various hyperparameters to improve the model performance. See Table 7 for the different parameters that were tried out. In addition, 3 different seeds were used to generate the dataset, and 4 different seeds for training the model. Not all possible combinations of hyperparameters were run. The result shown in the paper uses 0 layers, a batch size of 25,6 with 100 epochs and a gradient clipping of 0.5 during training.

Table 7: **Hyperparameter Values Tried Out for the Galaxy Experiment.** Epochs: number of epochs. Neurons: number of neurons per layer. Layers: number of layers of the NN. Batch Size: batch size during training. Grad. Clip: Gradient clipping during training.

| Epochs | Neurons | Layers | Batch Size | Grad. Clip. |
|---|---|---|---|---|
| 50, 100 | 50, 75, 100 | 0, 1, 2, 3 | 32, 64, 128, 256, 1024, 2048 | 0, 0.5, 1, 2 |

### D.2 Model Comparison Details

Data for these experiments is synthetically generated by first sampling an acyclic causal graph with the specified number of nodes and edges. We create a causal Bayesian network model for this graph, adding two additional nodes for each original node: the respective counterfactual node and an exogenous variable. The counterfactual nodes follow the same graph structure as their original twins, mimicking the "twin-network" (Balke & Pearl, 1994) approach. Any exogenous variable is connected to be a parent of both the respective original and counterfactual world node, but not any other node (we do not consider unobserved confounding). We generate random conditional probabilities for all endogenous variables, but then move all randomness to the exogenous variables only, such that each node value can be perfectly and uniquely determined if all parent variables (including the exogenous variable) are known. The value of any exogenous variable is determined as follows. First, an exogenous variable is thought to be sampled as a uniform value between 0 and 1. The value of the endogenous variable is 1 if and only if this sampled value is smaller than or equal to the conditional probability under which the variable would be 1 given its endogenous parents. This approach allows us to consider equivalence classes for the exogenous variables that produce the same results for any possible setting of endogenous parent variables. For example, an endogenous node with one parent variable would have 2 conditional probabilities given its parent. This would result in 3 equivalence classes for the exogenous variable, namely the values where the uniform samples are smaller than any conditional probability (node takes value 1), one where uniform samples are larger than one, but smaller than the other conditional probability (node takes value 1 or 0 depending on the parent value), and one where uniform samples are larger than any conditional probability (node takes value 0). The range of these equivalent samples is then used as the prior probability of the discrete value that the exogenous variable can take. As the possible exogenous variable settings explode exponentially with the number of endogenous parent nodes,

we cap the possible exogenous values at 10, ensuring that any two exogenous values result in different values for at least one setting of parent variables.[14] This way, all randomness is contained within the exogenous variables, allowing us to use this CBN both for generating observational and counterfactual samples, and for computing ground truth counterfactual probabilities, which we can compare with CF-SPN and CNF predictions in our evaluation. For each edge-node configuration and seed, 10,000 samples are generated for each possible counterfactual intervention and the no-intervention case (where the original and counterfactual worlds are identical). The cf-SPN is trained on this dataset of counterfactual samples for 100 epochs. The CNF is trained on the observational data and the causal graph.

For the evaluation, 100 random but unique original world and intervention combinations are sampled (such that each evaluation is different), each representing a different counterfactual query.[15] We employ two different strategies, depending on the problem size. For the four smaller configurations (up to 20 nodes and 30 edges), these counterfactual queries are sampled randomly from the space of all possible queries (avoiding repeating queries). These queries require the CBN to compute the probabilities of the counterfactual world by considering the possible exogenous variables compatible with the original world. However, this requires an exponential amount of both time and storage, such that it was impossible for us to do the same computation using CBNs for the larger problems. Therefore, we use sampled queries from a separate test set (compared to the other case, queries are now not sampled uniformly from the entire possible space but instead sampled according to the probabilities of the CBN), which allows us to provide exogenous values for the CBN, making computation feasible. However, this has two downsides. For one, the inference time we record for these configurations for CBNs is lower than it would be if they had to compute the queries correctly. Secondly, the resulting counterfactual probabilities change, as, instead of estimating the possible exogenous values ("abduction", in Pearl's three-step procedure), only one specific value of the sample is provided, even when other values would be possible given the original world setting. In any scenario, we then put the respective query (with or without specific exogenous values) into the CBN, which determines the marginal probability of each variable using variable elimination. We use these probabilities as gold standard probabilities and then compare the cf-SPN and CNF outputs to those.

For the cf-SPNs, we can also compute the marginal probabilities for each variable directly. For CNFs, on the other hand, we only compute a counterfactual sample. In many cases, such as for our experiments in Sections 4.1 and 4.2, counterfactuals can be determined fully deterministically. In this case, the NF might output exactly this counterfactual world correctly, which also fits the marginal probabilities that are 1 or 0 as well. However, since the exogenous variables can not be inferred precisely every time, the true counterfactual probability could also not be 0 and 1, in which case the NF might still rather predict the most likely sample, i.e., 0 or 1. Our evaluation considers multiple metrics to take this into account. First, we interpret the NF output as probabilities, so a sample of 1 would be a probability of 100% for the positive class. We then calculate the difference between the true marginal probability and the model (cf-SPN, CNF) probability and record the L1 and L2 error. Note that this is not entirely fair, as the CNF outputs technically do not represent probabilities. On the other hand, the CNF could theoretically also predict samples between 0 and 1, which is why interpreting these outputs as marginal probabilities is the best option, considering their general inability to truly compute marginal probabilities. Additionally, we record two other metrics that do not suffer from this problem. We compute the per-sample and element-wise accuracy for each counterfactual by mapping all (marginal) probabilities to 1 that are larger than 0.5 and to 0 otherwise. We then compare the CBN output with the cf-SPN and CNF and either count the sample as correct if all variable values match (per-sample) or record each correct and incorrect variable value separately (element-wise). By definition, the element-wise accuracy is likely to be much higher than the per-sample accuracy.

The entire evaluation is conducted over the following node/edge configurations: 5/5, 10/12, 15/20, 20/30, 50/100, 100/250. Each configuration is run for five different seeds, changing not only the model training, but also generating a new and different causal model and data each time.

---

[14]Not doing so would make computation in the Bayesian Network infeasible to run due to exponentially increasing space and time constraints.

[15]Any counterfactual intervention must change the value of the respective original world variable, so interventions that do not change the variable state are not considered.

| Seed | 5-5 | 10-12 | 15-20 | 20-30 | 50-100 | 100-250 |
|---|---|---|---|---|---|---|
| Time (s) | 66.69 ±1.78 | 235.74 ±16.30 | 522.15 ±21.27 | 951.03 ±57.02 | 4394.05 ±231.03 | 19700.45 ±672.77 |

Table 8: **cf-SPN Training Times (in seconds).** Training ranges from around a minute for the smallest problem to around 5.5 hours for the largest.

# E  Further Experimental Results

We include additional experimental results on tabular data for iSPN (Appendix E.1) and cf-SPN (Appendices E.2 and E.3), the comparison between different models (Appendix E.4), and the galaxy experiment (Appendix E.5).

## E.1  Tabular Data: iSPN

We apply an iSPN on the deterministic problem depicted by the causal graph in Figure 7. The approximated distribution for an iSPN trained on data generated from this setup is shown in Figure 10. As in the main paper, we specify an intervention (or no intervention) and calculate the probabilities for each variable. The ground truth is shown in bars, and the model prediction is shown as continuous, green lines (as Gaussian distributions are used in the SPN leaves). The presented results show the average of 1000 samples. In this experiment, the iSPN learns both the observational and the interventional distribution very well. For further experiments on iSPNs, we refer the reader to Zečević et al. (2021).

## E.2  Tabular Data: cf-SPN

Consider again the causal graph from Figure 7 in its deterministic form. The results in Figure 11 show the probabilities on the problem when uniformly, randomly intervening on a variable. It can be seen that the model matches the ground truth without any visible error.

The non-deterministic case has been discussed in the main paper (Section 4.2). Figure 12 shows results for interventions on variables not shown in the main paper, highlighting the good model performance throughout.

## E.3  Watering: cf-SPN

The watering example has been discussed in detail in the main paper (Section 4.1). Figure 13 provides additional results.

## E.4  Model Comparison

Training times for the cf-SPN are shown in Table 8. In these experiments, all cf-SPNs are trained for 100 epochs, resulting in training times between not much more than one minute for the smallest problems and around 5.5 hours for the largest problems. Since no early stopping is used, the training loss often converges much earlier.

In the evaluations, there were rare instances where an evaluation had to be discarded because the probability of the data point in the CBN was 0. This led to the CBN returning no valid values to probabilities we could compare to. We omitted these instances, as they were especially rare. In our configurations, 1 seed in the 10-node configuration, 2 seeds in the 15-node configuration, and 2 seeds in the 20-node configuration were affected. In the first two configurations, this affected only a single evaluation out of all 100, while in the 20-node case, 5 and 4 out of 100 evaluations each were affected. Also note that for 3 seeds in the 5-node configuration, all samples were ID. Therefore, the mean and standard deviation for the OOD row were only computed using a few evaluations on these two seeds.

Tables 9 and 10 provide two additional metrics for the experiments in Section 4.5. The L1-error shows that CNFs generally perform worse than cf-SPNs when it comes to computing probabilities, even if the difference is not as stark as for the L2-error. For the accuracy, we see cf-SPN starting better than CNF in ID regimes,

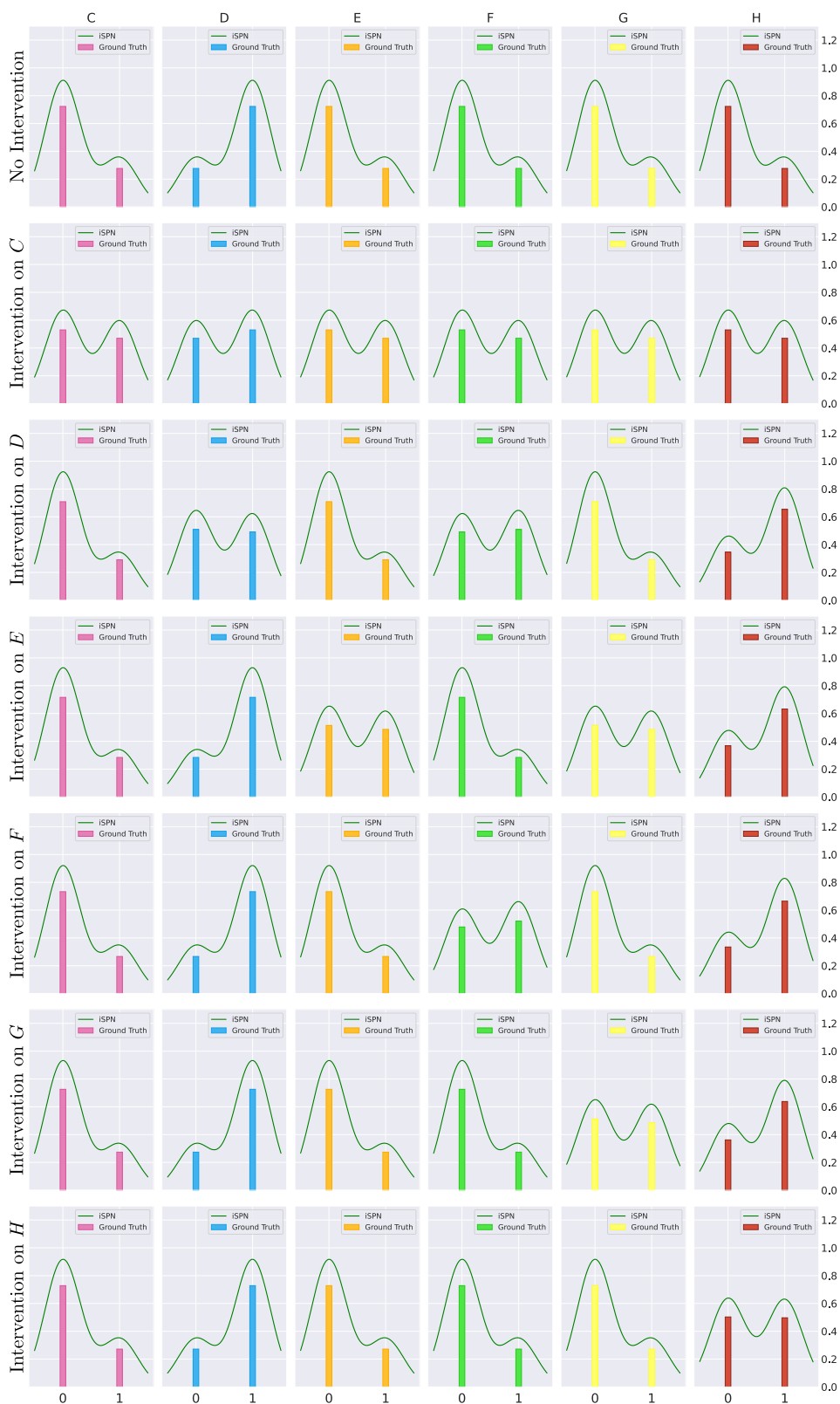

Figure 10: **iSPN Full Results Experiment.** Showing iSPN predictions (green lines, probability density functions) and the ground truth (bars, probability mass functions). All graphs were created by sampling 1000 instances from the causal model. (Best viewed in color.)

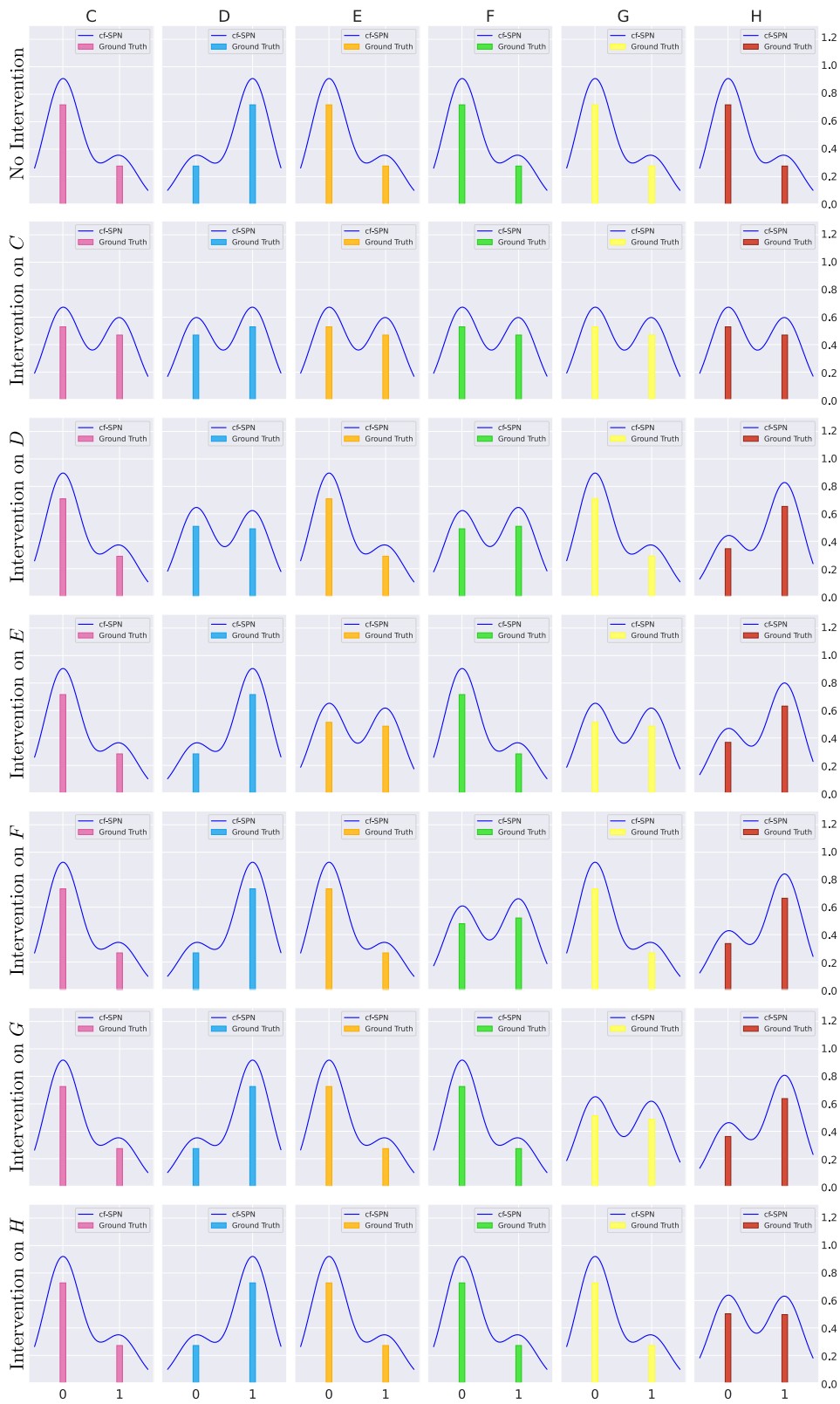

Figure 11: **cf-SPN Full Results Experiment without Noise.** Showing cf-SPN predictions (blue lines) and the ground truth (bars). All graphs were created by sampling 1000 instances from the causal model. (Best viewed in color.)

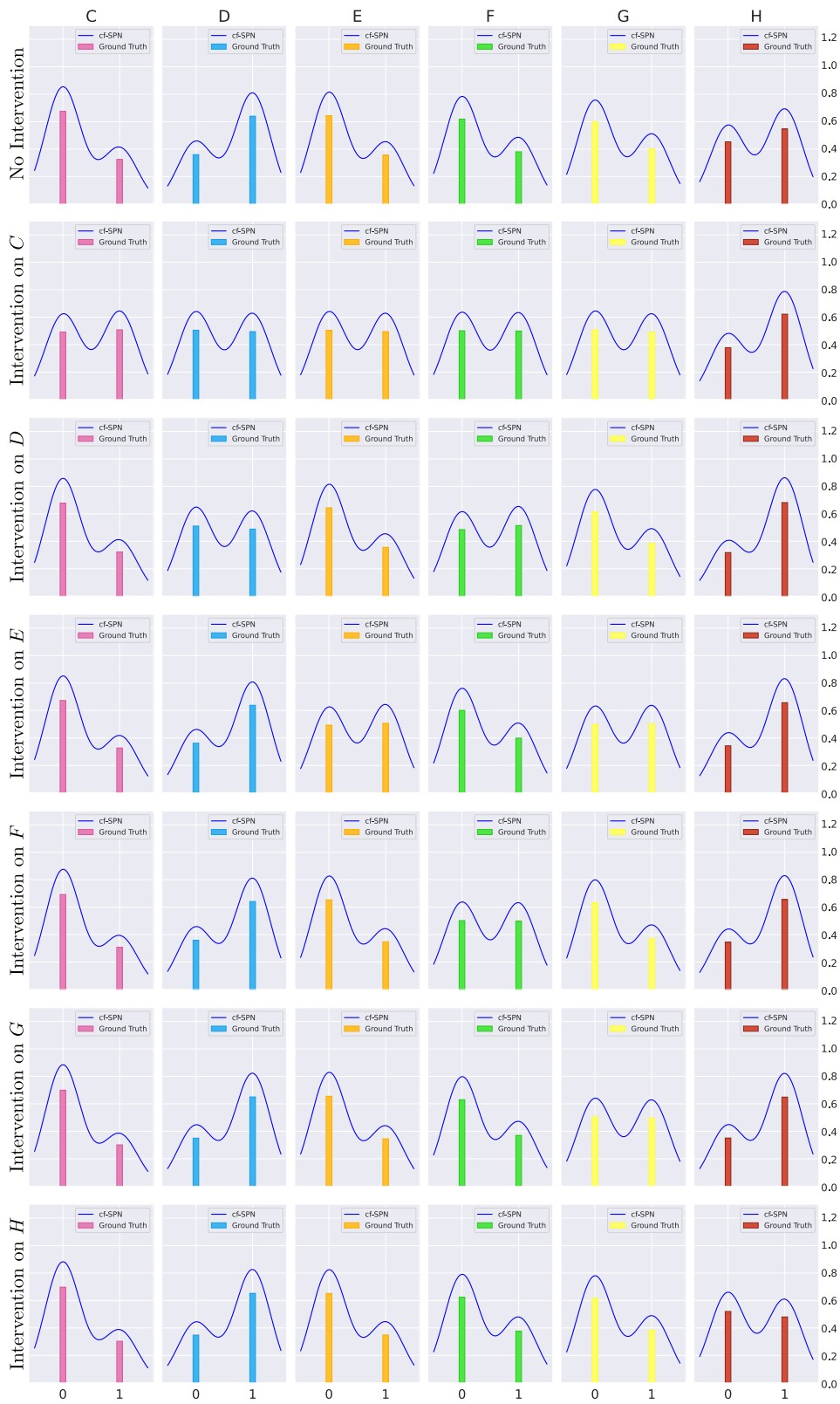

Figure 12: **cf-SPN Full Results Experiment with Noise.** Showing cf-SPN predictions (blue lines) and the ground truth (bars). All graphs were created by sampling 1000 instances from the causal model. (Best viewed in color.)

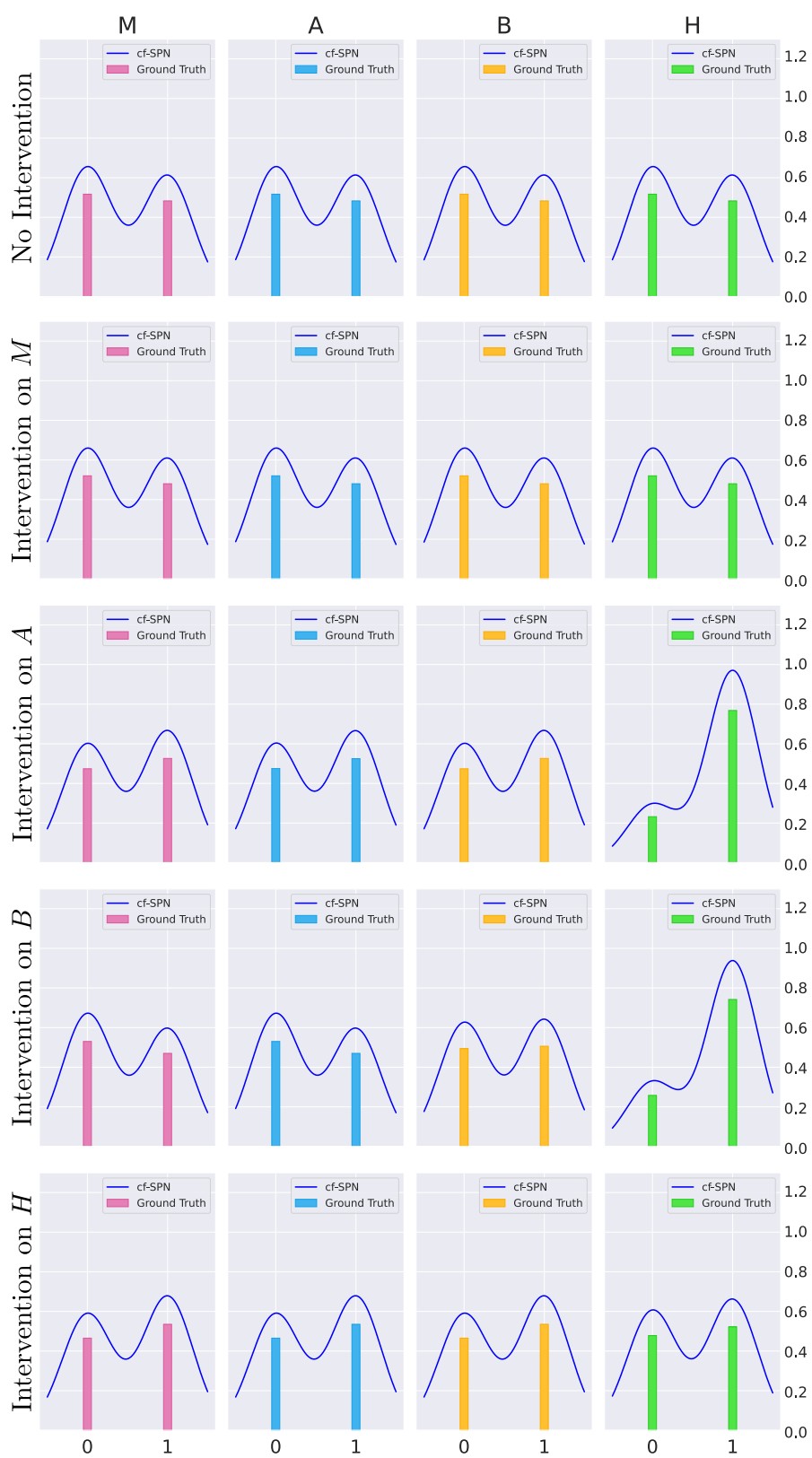

Figure 13: **cf-SPN Full Watering Experiment Results.** Showing cf-SPN predictions (blue lines) and the ground truth (bars). All graphs were created by sampling 1000 instances from the causal model. (Best viewed in color.)

|  | Method | 5/5 | 10/12 | 15/20 | 20/30 | 50/100 | 100/250 |
|---|---|---|---|---|---|---|---|
| ID | cf-SPN | $1.79 \pm 0.47$ | $1.96 \pm 0.61$ | $2.88 \pm 1.67$ | N/A | N/A | N/A |
|  | CNF | $6.81 \pm 2.75$ | $3.50 \pm 1.18$ | $2.63 \pm 1.77$ | N/A | N/A | N/A |
| OOD | cf-SPN | $5.02 \pm 1.28$ | $2.35 \pm 0.65$ | $2.19 \pm 0.44$ | $2.18 \pm 0.27$ | $2.27 \pm 0.11$ | $3.28 \pm 0.32$ |
|  | CNF | $10.97 \pm 0.23$ | $5.30 \pm 0.87$ | $4.52 \pm 1.18$ | $3.53 \pm 0.71$ | $1.81 \pm 0.09$ | $1.33 \pm 0.30$ |

Table 9: **L1-Errors of Marginal Probabilities for cf-SPNs and CNFs (scaled by 100).** Different combinations of nodes/edges are evaluated, ranging from 5 to 100 nodes. Some counterfactual queries have been part of the training data (ID), others have not (OOD). The table shows the average L1-error for the marginal probability per variable times 100 (as most differences are quite small). In-distribution samples are predicted with lower errors, and cf-SPNs give lower errors throughout most settings.

|  | Method | 5/5 | 10/12 | 15/20 | 20/30 | 50/100 | 100/250 |
|---|---|---|---|---|---|---|---|
| ID | cf-SPN | $0.97 \pm 0.03$ | $0.92 \pm 0.05$ | $0.66 \pm 0.19$ | N/A | N/A | N/A |
|  | CNF | $0.91 \pm 0.03$ | $0.90 \pm 0.06$ | $0.70 \pm 0.17$ | N/A | N/A | N/A |
| OOD | cf-SPN | $0.84 \pm 0.09$ | $0.86 \pm 0.07$ | $0.80 \pm 0.05$ | $0.77 \pm 0.03$ | $0.64 \pm 0.04$ | $0.12(*) \pm 0.02$ |
|  | CNF | $0.94 \pm 0.06$ | $0.91 \pm 0.04$ | $0.81 \pm 0.04$ | $0.79 \pm 0.03$ | $0.63 \pm 0.01$ | $0.50 \pm 0.05$ |

Table 10: **Accuracy Comparing cf-SPNs and CNFs.** Different combinations of nodes/edges are evaluated, ranging from 5 to 100 nodes. Some counterfactual queries have been part of the training data (ID), others have not (OOD). Generally, CNFs tend to perform slightly better on OOD samples, but slightly worse on ID samples. Even in the OOD cases, cf-SPNs do not fall far behind, except for the 100/250 configuration (* more information in the text).

where the model memorizes the training data well. On the other hand, the CNF's knowledge of the correct causal graph can help with predicting OOD samples correctly, resulting in better results there. Notably, the accuracy for 100 nodes and 250 edges for cf-SPNs at 0.12 is surprising. This has several explanations. For one, we did not conduct any hyperparameter optimization in these experiments. As we also explained for the galaxy collision experiment, hyperparameter optimization can be quite important, and a bad choice of parameters can lead to bad results. Additionally, the explicit integration of the causal graph for CNFs puts it at an advantage here, where it is impossible for the CNF to incorrectly predict counterfactual world variables that are not affected by the counterfactual intervention, while any one of the 100 variables could be incorrectly predicted by the cf-SPN, reducing sample-wise accuracy. Still, the other metrics show that the cf-SPN performance is far better than random, and an optimized choice of hyperparameters would probably again lead to better results.

### E.5 Galaxies

In Figure 14, we show a more extensive illustration of the galaxy experiment introduced in the main paper (Section 4.4). Here, the displacement of the particles to lower positions after time step 49 can be observed. While the model performance here shows visible errors compared to the ground truth simulation, the model still shows sensible results despite more than 70 time/prediction steps that can all introduce new errors.

## F Technical Details

All models were built using PyTorch and trained and evaluated in the following environment: Intel(R) Xeon(R) Platinum 8174 CPU @ 3.10GHz, RAM: 1546 GB, GPU: NVIDIA Tesla V100-SXM3-32GB with 32 GB of RAM.

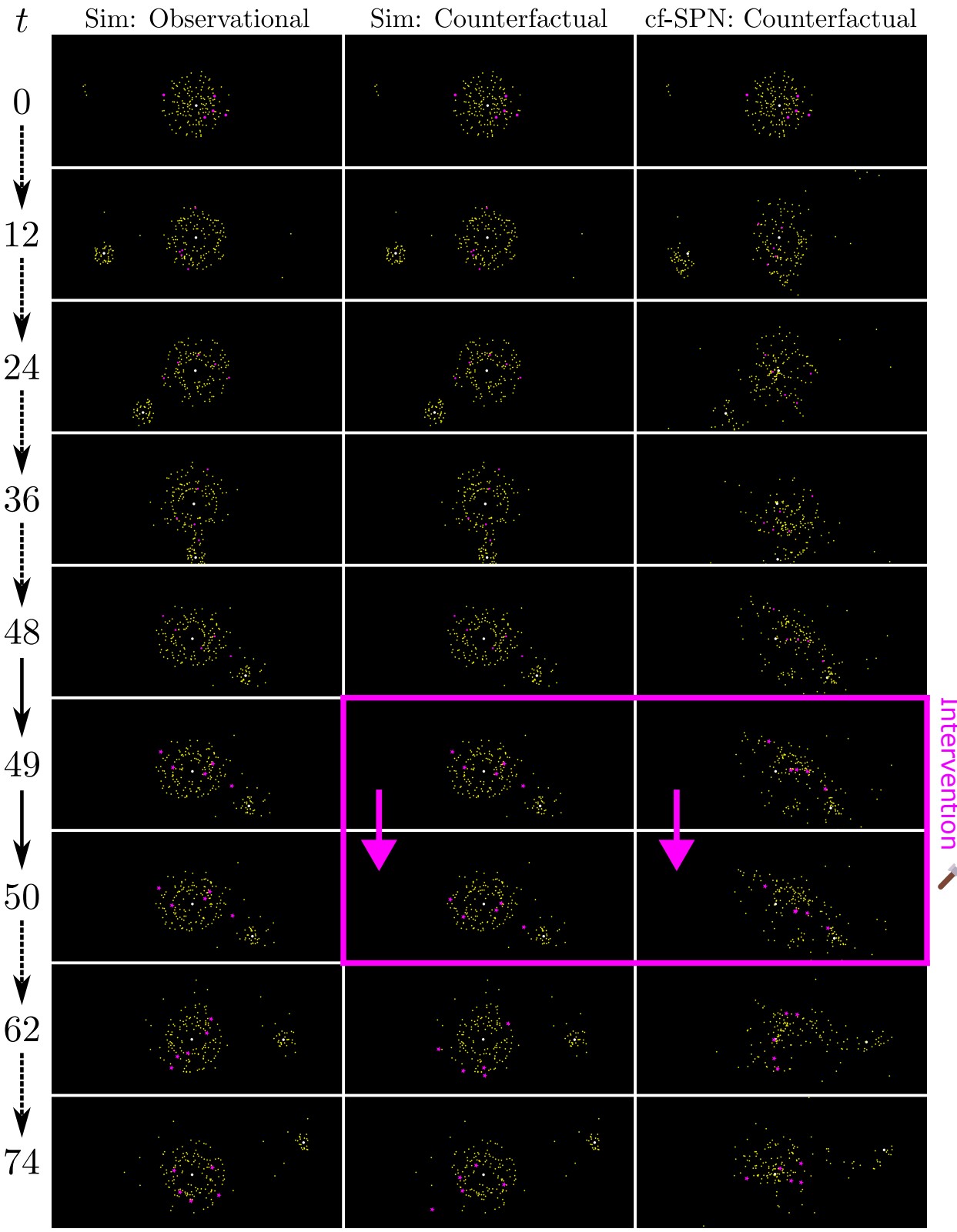

Figure 14: **cf-SPN Galaxies Experiment.** Sim: Simulation. The interventions take place from time step 49 to time step 50 on the magenta particles. From time step 50 onward, the visual shapes of these particles are additionally changed to stars. (Best viewed in color.)

