# OpenReview forum: "Structural Causal Circuits: Probabilistic Circuits Climbing All Rungs of Pearl's Ladder of Causation"
_TMLR — Accepted by TMLR_

### Review · Reviewer_b7eJ · 2025-06-08

**Summary Of Contributions:**

- An extension of SPNs for counterfactual estimation.
- A new class of models "SCCs", merging existing classes of SPNs (observations, interventional and their counterfactual variant).

**Audience:**

Yes

**Broader Impact Concerns:**

-

**Claims And Evidence:**

Yes

**Requested Changes:**

- I'd like the authors to clarify the following issues.
- I'm not sure I fully understood this sentence: "Previous work on inference using tractable circuits shows that exact computation of interventional distributions is #P-hard (Wang & Kwiatkowska, 2023). There certainly are cases where this can cause problems for SCCs."
    - What kind of "problems" are the authors referring to?
- Following up: "However, we argue that most practical applications feature compressible distributions that can be represented by circuits, thereby benefiting from tractable inference."
    - What are the "compressible distributions"?
    - Is this the class of distributions that can be represented by SCCs?
    - In which way they are different from distributions represented by CBNs?
- Moreover: "We, therefore, relax the equality constraint and say that our models should “truthfully approximate” the probability distribution up to some small ϵ."
    - The "approximation" part is related to some "distance up to epsilon" in the parameter space?
- Often authors say that an "SPN is learned using data".
    - Do they refer to the parameters?
    - Or also the structure?
    - In which way one can learn an SPN from data?

**Strengths And Weaknesses:**

- Strengths:
    - Linear time computation for counterfactual queries using SPNs.
    - Formal proof of correctness w.r.t. language framework provided by Bareinboim for causal inference.
    - Wide range of experiments to show how to use these models in practice.
    - Public code for reproducibility.

- Weaknesses:
    - There are multiple instances in the text in which authors are vague about the actual learning procedure, e.g. "[it] is unreasonable to expect any model to learn the 'true' SCM for a specific problem", but then also "domain experts could potentially set up models with the exact weights, yielding a model that matches the causal distribution perfectly". I think there is some confusion about the actual specification of a model from prior knowledge and the learning procedure from limited data, I would prefer to read about these points in a dedicated section rather than having them sparse here and there, it makes me wonder whether the authors deals with one setting or the other, or both.
    - Lack of comparison with standard de-facto CBNs. Authors state that exact inference in CBNs is np-hard and that the only tractable inference is approximate. Therefore, to have "exact inference" one must resort to SCCs. However, there is no comparison to between SCCs and CBNs, which is quite unexpected since it may be the case that approximate inference in CBNs would result in a good approximation.
    - While this representation sounds promising I find it hard to use these models for counterfactual query since their structure is designed for computation rather than interpretation of the semantics of the system itself.

---

> ### Author Response · Authors · 2025-07-07
> **Rebuttal (1/2)**
>
> We thank the reviewer for their valuable feedback. Based on this, we made several changes to our paper to clarify some points and provide additional explanations and experimental evidence. We used the following colors to highlight changes in our revision: Red (relevant for all reviewers), blue (Reviewer QSki), green (Reviewer i4VJ), and orange (Reviewer b7eJ).
>
> We now respond to the weaknesses (W) and requested changes (C):
>
> **W1: Learning Process.**
> When we mention the "true" SCM for a specific model, we are referring to the fact that multiple SCMs can generate the same data, especially on lower rungs. Therefore, a single "true" model can not always be expected to be learned. Setting up a perfect expert-engineered SCC is not trivial, but knowledge about clusters of data (sum nodes) and context-specific independencies (product nodes) could certainly help. We do not propose a detailed approach on how to achieve this, but merely highlight the idea and the potential. More easily, expert knowledge could provide an SCM or counterfactual data, which can then be used to train a model.
> To answer the reviewer's point more generally: Learning, in the context of SCCs, refers to the learning of the distributions as described by the data. This includes interventional and counterfactual distributions for $SCC_2$ and $SCC_3$, where each such distribution is then represented by the SPN after the NN set the SPN parameters according to the input.
>
> **W2: Comparison to CBNs.**
> We added a new experiment where we compare cf-SPNs to Causal Bayesian Networks (CBNs) and Causal Normalizing Flows (CNFs) in Section 4.5. While we use the CBNs as a gold standard when determining errors, we can see that the cf-SPN is quite close to the CBN predictions. Additionally, inference times of both models start off similarly, but cf-SPNs only scale roughly linearly in the number of variables in our experiments, while the inference time for CBNs explodes!
>
> (**Please note**: Due to the #P-hard complexity of computing marginal probabilities within CBN, only three out of 5 seeds for the 100 node configuration have finished as of now. Therefore, the 100 node results are only determined by 3 seeds. We will update results once the other two seeds finished computing, which we expect to be the case in about 2 days. For the time being, our revision consists of the three seeds that have finished in time.)
>
> **W3: Computation rather than Semantics.**
> SPNs, and therefore SCCs, trade the 'semantic' interpretability of SCMs, for learning a compressible partition function that allows for tractable inference in the number of nodes; a property that can not be obtained within Bayesian Networks (where marginal inference is #P-hard). The presentations learned by SCCs do not represent a semantic structure similar to that of structural causal models, but rather present an efficient representation of the distributions' partition function [1]. While this might complicate validation of the learned mechanisms, our experiments show good coherence with the underlying ground truth data. As with all approaches that rely on learning mechanisms purely from data, adequate amounts of data with sufficient variability are required for the model to identify the underlying mechanisms and converge to the true distribution. For the individual case, corresponding testing and validations should be put in place to secure the safe operation of the model. Thank you for raising this important point again! We have added a 'limitations' paragraph within the conclusion to better highlight the possible limitations of our work.

---

> > ### Author Response · Authors · 2025-07-07
> > **Rebuttal (2/2)**
> >
> > **C1: Problems for SCCs.**
> > We agree that this sentence was confusing. We changed the paragraph in the paper (start of Section 3, pages 6/7). To still answer the question: Since the problem is fundamentally #P-hard, there exist interventional distributions that can not be computed exactly. However, we argue that most practical distributions are compressible (see next point) and, therefore, (context-specific) independencies in the data can be leveraged to still compute causal queries efficiently. Note that inference in SCCs will always be tractable, but in the worst case, the predicted probabilities could be approximations and not exact. Increasing the model size can then improve the quality of the approximation.
> >
> > **C2: What Distributions can a SCC Represent?**
> > Increasingly larger SCCs can approximate increasingly complex distributions arbitrarily well. In cases where distributions are sufficiently compressible, i.e., they contain many (context-specific) independencies, even smaller SCCs will be able to learn them exactly and efficiently. For increasingly complex distributions, larger structures can be used to approximate them with increasing precision. We added a new clarification on this in Section 3, along with the changes mentioned for C1.
> >
> > **C3: Approximation in Parameter Space.**
> > For some distributions, an SPN of limited size might be insufficient to represent the distribution exactly. However, we do know that increasing the SPN size (adding parameters) can also improve any such approximation. Since we are dealing with limited-size SPNs, this means that a "distance up to $\epsilon$" between the true distribution and the modeled distribution has to be expected.
> >
> > **C4: What Parameters are Learned?**
> > In this paper, we use a random SPN architecture [2], which sets the architecture, so the architecture is given when we start learning the model. We learn the parameters, i.e., the distributions in the leaves of the SPN, the sum weights in the SPN, and the NN weights. There is also related work on learning the architecture (e.g., [3]), but this is not done in this paper.
> >
> > ---
> >
> > We hope the new experiments and our answers have helped clarify our work. Please let us know if any questions remain!
> >
> > Best regards,
> > the authors
> >
> > [1] Martens, James, and Venkatesh Medabalimi. "On the expressive efficiency of sum product networks." arXiv preprint arXiv:1411.7717 (2014).
> >
> > [2] Peharz, Robert, et al. "Random sum-product networks: A simple and effective approach to probabilistic deep learning." *Uncertainty in Artificial Intelligence*. PMLR, 2020.
> >
> > [3] Gens, Robert, and Domingos Pedro. "Learning the structure of sum-product networks." *International conference on machine learning*. PMLR, 2013.

---

### Review · Reviewer_i4VJ · 2025-06-09

**Summary Of Contributions:**

The paper introduces Structural Causal Circuits (SCCs), a family of Sum-Product Networks (SPN) that is capable of answering observational, interventional and counterfactual queries. In particular, counterfactual SPN (cf-SPN) is its key contribution. The main idea is to feed both the mutilated graph that encodes the intervention and the original-world assignment into a neural network that parameterises a standard SPN. Thus, by maximizing the log-likelihood supervised by the counterfactual data, the cf-SPN learns to answer rung 3 queries. The author showed that the proposed method can identify the counterfactual distribution and evaluated its performance on synthetic and physical-based datasets.

**Audience:**

Yes

**Claims And Evidence:**

Yes

**Requested Changes:**

1. Explanations on the effect of noise distributions in the leaf node in modelling data distributions.
2. Include a discussion on the training time.
3. Provide experiments on counterfactual predictions evaluated by a quantitative metric.
4. If possible, investigate the robustness of the cf-SPN trained on approximated counterfactual data.

**Strengths And Weaknesses:**

**Strength**
1. The paper is fairly well-written, with motivation and preliminaries of SPN clearly discussed.
2. The proposed method provides exact inference for rung 1,2,3 queries in tractable time. Existing methods either are not tractable or can not answer all kinds of queries.
3. Experiments showed promising performance of the proposed method on real-world datasets.
4. The additional experiments on unseen inputs are especially convincing. The results demonstrate that cf-SPN generalizes well to unseen counterfactuals when it is only trained on a limited number of input data and interventions.

**Weakness**
1. The method requires knowledge of counterfactual data in the training, which in many cases is impossible. While the author has discussed some ideas to acquire/estimate counterfactual data, it is not clear whether the proposed method is robust to such approximated data.
2. It is not clear to me whether the choice of the distributions in the leaf node would affect the conclusion in Proposition 3. For example, by choosing the Gaussian distribution, the predicted distributions by the SPN in Figure 4-6 seem to be Gaussian mixtures, which are continuous. However, the true distribution/pointwise prediction should be discrete. I am a bit confused about interpreting these continuous distributions.
3. I think the benefit of exact tractable inference in cf-SPN would be more strengthened if the author could compare it with other methods, e.g., Causal NF, on counterfactual predictions evaluated by RMSE, see section 6.1 of the Causal NF paper.
4. While the SPN provides tractable inference, the paper did not discuss the computational burden required for training and how it scales with the number of variables.

**Minor**
1. The last sentence on Page 7 seems to be incomplete. References required.
2. Section 4.1, first sentence incomplete.

---

> ### Author Response · Authors · 2025-07-07
> **Rebuttal (1/2)**
>
> We thank the reviewer for their valuable feedback. Based on this, we made several changes to our paper to clarify some points and provide additional explanations and experimental evidence. We used the following colors to highlight changes in our revision: Red (relevant for all reviewers), blue (Reviewer QSki), green (Reviewer i4VJ), and orange (Reviewer b7eJ).
>
> We now respond to the weaknesses (W), as well as requested changes (C), and minor points (M):
>
> **W1, C4: Approximated Counterfactual Data.**
> The cf-SPN learns the data distribution as provided by the given dataset. If the provided data is approximate or noisy, the model will learn this uncertainty as part of the probability distribution. For example, in one experiment in Section 4.6, the functions are inherently noisy (predicting the opposite class 10% of the time). This is then learned by the cf-SPN, which results in different distributions than if the noise would not have been present. On the other hand, if approximate data is not just noisy (as in Gaussian noise) but actively faulty, then the cf-SPN will learn just that. While this results in a cf-SPN that does not fully represent the "true" distribution, it can only learn what is provided by the data. In a sense, the data is the ground truth for the cf-SPN. The challenge of obtaining good counterfactual data is, therefore, separate from the learning task of the cf-SPN. We added a paragraph elaborating on this in Appendix C.
>
> **W2, C1: Choice of Leaf Distributions.**
> We use Gaussian leaves for their general versatility in modeling both continuous and binary leaves. In our experiments on binary data, we, therefore, end up with probability densities (see Figures 4, 5, 6) where we can easily extract probabilities for 0 and 1 by predicting the density for both values and normalizing. It is possible to directly use binary distributions in the leaves, but this was not necessary for our experiments, where continuous leaves can also model these distributions, as well as continuous distributions (as we use in the particle collision and galaxy experiments). With respect to Proposition 3, this only matters for the last part of the proof, stating that SPNs can learn joint probability distributions. While here, binary leaves would provide a "cleaner" approach for modeling binary problems, continuous leaves are also capable of doing so, as explained above. We added a short explanation for the usage of Gaussian leaves in Appendix D.
>
> **W3, C3: Experimental Evidence for Tractable Inference and Quantitative Evaluation.**
> We added a new experiment where we compare cf-SPNs to Causal Bayesian Networks (CBNs) and Causal Normalizing Flows (CNFs) in Section 4.5. When determining predictive accuracy, we use CBNs as a gold standard baseline. We include quantitative evaluations for using different metrics. Our experiments show that the cf-SPNs are quite close to predictions provided by CBNs (Table 2), especially when measuring the error between marginal probabilities (Table 3). CNFs give slightly higher accuracies than CBNs (Table 2), but they require the causal graph to be known and are unable to perform marginalization. By using their predictions and interpreting them as marginal probabilities, they perform worse than cf-SPNs (Table 3). When considering the inference times for the computation of marginal probabilities, we see that cf-SPN are close to CBNs for smaller problems, but their tractability allows them to make predictions much more efficiently for increasingly large problem sizes (Table 5).
>
> (**Please note**: Due to the #P-hard complexity of computing marginal probabilities within CBN, only three out of 5 seeds for the 100 node configuration have finished as of now. Therefore, the 100 node results are only determined by 3 seeds. We will update results once the other two seeds finished computing, which we expect to be the case in about 2 days. For the time being, our revision consists of the three seeds that have finished in time.)

---

> > ### Author Response · Authors · 2025-07-07
> > **Rebuttal (2/2)**
> >
> > **C2, W4: Training Time of cf-SPNs.**
> > In our new experiments, we trained all cf-SPNs for 100 epochs. In the smallest case (5 variables), training, depending on the seed, took between 49.66 and 60.15 seconds, in the largest case (100 variables), between 17669.74 and 18548.17 seconds (around 5 hours). This increase in training time is not only because of the larger variable size but also because we use more data to train on the larger problem, as small problems do not require as much data for the data domain to be sufficiently represented. Even on the big problem, we see little change in loss after 40 out of the 100 epochs, and even after 20 epochs, the loss is already almost converged. We added information on training times in Appendix E.4.
> >
> > The main advantage of the cf-SPN is tractable inference. In lots of applications, one might be ok with spending a larger amount of time training *once* in order to be able to perform fast inference afterward: here, inference speed matters much more.
> > With respect to both training and inference, also note that there are other approaches that might accelerate SPN speed even further and were not used here [1,2].
> >
> >
> > ---
> >
> > We hope the new experiments and our answers have helped clarify our work. Please let us know if any questions remain!
> >
> > Best regards,
> > the authors
> >
> >
> > [1] Peharz, Robert, et al. "Einsum networks: Fast and scalable learning of tractable probabilistic circuits." *International Conference on Machine Learning*. PMLR, 2020.
> >
> > [2] Liu, Anji, Kareem Ahmed, and Guy Van Den Broeck. "Scaling Tractable Probabilistic Circuits: A Systems Perspective." *International Conference on Machine Learning*. PMLR, 2024.
> >
> > **M1, M2: Minor Changes**
> > We added a reference on page 7 and slightly changed the first sentence in 4.1.

---

### Review · Reviewer_QSki · 2025-06-17

**Summary Of Contributions:**

The paper presents an extension of sum-product networks (SPNs) to model counterfactual distributions, unifying it with existing observational and interventional SPNs, This facilitates tractable inference at all three rungs of Pearl's causal ladder.

**Audience:**

Yes

**Claims And Evidence:**

No

**Requested Changes:**

**Crucial:**
1. Claims about in/tractability need some more support and
   clarification. The only concrete claim is that the complexity of
   inference in SPNs is #P-hard. Can the authors say anything more
   specific? How does this compare to complexity of inference in
   Bayesian networks? Including experiments comparing runtime for SPNs
   vs, say, MLE discrete DAG models would be very informative here.
2. In addition to runtime plots, it would be good a performance check to
   see how SPNs compare to, say, MLE discrete or Gaussian DAG models.
   The SPN performance seems quite good, but it would be helpful to
   know what simple DAG model performance is like as a baseline.
3. I'm not familiar with SPNs, and after reading this I'm still not
   sure if SPNs are universal approximators or not. Are SPNs just a
   clever, tractable way of modeling arbitrary distributions, or does
   the tractability come at a cost (essentially a parametric
   restriction on the class of distributions)? Some high-level
   discussion related to this would be helpful in Section 2.3.
4. The top of page 7 argues that "most practical applications feature
   compressible distributions that can be represented by circuits,
   thereby benefiting from tractable inference". What's the
   support/evidence for this argument?

**Suggested:**
1. Section 2.3 says the structure of an SPN is described by a DAG. Can
   it be any DAG, or e.g., only a tree?
2. Section 2.4 references both Papantonis & Belle as well as
   Papantonisa & Bellea. The latter seems to be a typo.
3. Propositions 1,2, and 3 essentially say SPN $\subseteq$ SCC (for
   each of the respective rungs). What about containment in the other
   direction $\supseteq$?
4. Do I understand Section 3.4 correctly: I can use some causal
   discovery method to get a DAG, and then the proposed method here
   can model the actual distributions corresponding to this causal
   DAG? And perhaps there's some efficient/performant joint SPN-causal
   discovery possible, but that's left for future work?

**Strengths And Weaknesses:**

**Strengths:**
1. generally well written, clearly explained, and easy to follow
2. valuable extension and unification of SPNs for causal modeling
3. proposed method performs well in experiments

**Weaknesses:**
1. vague claims about tractability, and no experimental evidence for it
2. no experimental results comparing to non-SPNs
3. expressivity limitations of SPNs not so clear (are they universal approximators?)
4. these models don't seem to allow, e.g., prediction of unseen
   interventional distributions in the same way an SCM does (or do I misunderstand?)

---

> ### Author Response · Authors · 2025-07-07
> **Rebuttal (1/2)**
>
> We thank the reviewer for their valuable feedback. Based on this, we made several changes to our paper to clarify some points and provide additional explanations and experimental evidence. We used the following colors to highlight changes in our revision: Red (relevant for all reviewers), blue (Reviewer QSki), green (Reviewer i4VJ), and orange (Reviewer b7eJ).
>
> We now respond to the weaknesses (W), as well as crucial (C) and suggested (S) changes:
>
> **W1, W2, C1, C2: Comparison with DAG Models and Experimental Evidence for Tractability.**
> We added a new experiment where we compare cf-SPNs to Causal Bayesian Networks (CBNs) and Causal Normalizing Flows (CNFs) in Section 4.5. When determining predictive accuracy, we use CBNs as a gold standard baseline. Our experiments show that the cf-SPNs are quite close to predictions provided by CBNs (Table 2), especially when measuring the error of marginal probabilities (Table 3). CNFs give slightly higher accuracies than CBNs (Table 2). Note, however, that they require the causal graph to be known and are unable to perform marginalization to obtain this performance. By using their predictions and interpreting them as marginal probabilities, they perform worse than cf-SPNs (Table 3). When considering the inference times for the computation of marginal probabilities, we see that cf-SPNs are close to CBNs for smaller problems, but their tractability allows them to make predictions much more efficiently for increasingly large problem sizes (Table 5).
>
> (**Please note**: Due to the #P-hard complexity of computing marginal probabilities within CBN, only three out of 5 seeds for the 100 node configuration have finished as of now. Therefore, the 100 node results are only determined by 3 seeds. We will update results once the other two seeds finished computing, which we expect to be the case in about 2 days. For the time being, our revision consists of the three seeds that have finished in time.)
>
> **W3, C3: Does Tractability Come at a Cost in Terms of Expressivity?**
> SPNs can fundamentally model arbitrary probability distributions. There is no inductive bias on the types of distributions that can be learned. If distributions are compressible (see C4), then even smaller SPNs can learn or approximate them very well, while incompressible distributions might require larger SPNs. In full generality, an SPN of fixed size is not capable of modelling any distribution, but increasingly larger SPNs can approximate increasingly complex distributions arbitrarily well. We changed the corresponding paragraph in Section 3 to clarify this and also added further explanation on the expressivity-tractability tradeoff in Appendix A.
>
> **C4: Do most Practical Applications Feature Compressible Distributions?**
> Any (context-specific) independency present in the data distribution contributes to its compressibility. Since, especially large-scale graphs are usually sparse, we think it reasonable to assume that most practical applications are well compressible.
>
> **W4: Prediction of Unseen Intervention.**
> Our approach for iSPN and cf-SPN does not require the causal graph to be known. However, this also means that interventions that have never been seen as part of the training data before can not always be predicted by the model, as it would not know which parameters are affected by this new input (intervention) that has never been observed before in training. (As with all ML approaches that learn causal relations from data, sufficient variability of the data is required for the relations to become identifiable to the SPN.) An approach that integrates some form of explicit causal discovery, such that some unseen intervention could also be predicted correctly, is left for future work. While we have no guarantee on generalization to unseen interventions, note that we included experiments on unseen intervention settings, where the model manages to predict a larger number of interventions correctly while only being trained on single intervention data (Section 4.6).

---

> > ### Author Response · Authors · 2025-07-07
> > **Rebuttal (2/2)**
> >
> > **S1: SPN Structure.**
> > The completeness and decomposability properties put some restrictions on how the graph of an SPN may look. While SPNs are often depicted as trees, this is not a strict requirement, as nodes could be a child of multiple other nodes in higher layers of the SPNs, breaking the tree structure. So while not all DAGs are valid SPNs, the SPN DAG does not have to be a tree. We added a sentence on how such a DAG could look in the footnote in Section 2.3.
> >
> > **S2: Typo.**
> > Thank you, we fixed the typo.
> >
> > **S3: Relationship SPNs and SCCs.**
> > Thank you for the interesting question! Generally, the propositions do not make any statement about the other direction. If we consider definition 6, we can see that an SCC must be able to represent an observational/interventional/counterfactual probability distribution of the respective rung. For an observational SPN $m_1$, we can always find a probability distribution such that $m_1 \in SCC_1$ holds. Therefore, all SPNs are in $SCC_1$. For iSPN (and cf-SPN), however, the NN component changes the situation. In theory, the NN parameters of such a model could, for example, predict an intervention like $do(A=1)$ such that the probability for $A=0$ is 1. By definition, this is still an iSPN, but it would not satisfy the equality of the model and SCM probabilities as required in Definition 6. Obviously, this iSPN would not be a very useful model, and this behavior would not be learned given high-quality training data, but it still serves as an example why the opposite direction $ \text{iSPN} \supseteq SCC_2$ (and $\text{cf-SPN} \supseteq SCC_3$) does not hold for iSPNs (and cf-SPN).
> >
> > **S4: Causal Discovery in SCCs.**
> > We do not use explicit causal discovery in our approach. We assume that, in order to learn the causal probability distributions, the model must somehow implicitly learn the causal edges, i.e., which interventions have an effect on which other variables, but at no point is an explicit causal graph represented.
> > Also note that SPNs try to learn compact partitions of the distribution (i.e., *sets* of joint variables jointly (sums) or their independent decomposition (products)), while SCMs model (in)dependencies between *individual* variables. As such, it has not yet been shown how to efficiently(*1) transfer SCM representations into SPN structures. We therefore focus on a data-only approach and leave a joint approach combining causal discovery and the SCC learning process to future work.
> > Our response on semantics to Reviewer b7eJ (W3) and the new limitations paragraph on page 18 could also be of interest regarding this question.
> >
> >
> > ---
> >
> > We hope the new experiments and our answers have helped clarify our work. Please let us know if any questions remain!
> >
> > Best regards,
> > the authors
> >
> >
> > (*1) Meaning, without exponential blow-up in the size of the SPN.

---

> ### Comment · Reviewer_QSki · 2025-07-16
>
> Thanks for the thorough rebuttal! It along with the revision address all of my original concerns.
>
> One final, minor comment: the caption for Table 4 has a typo: "Runties"

---

### Comment · Action_Editor_Jpdg · 2025-07-02

The authors requested an extension, which I would grant (and I am waiting for it to be added into the TMLR system). I think this is beneficial as there were quite some expeirments and I think this will help to improve the quality.

---

### Author Response · Authors · 2025-07-23
**Missing Experimental Results**

Dear Reviewers,

Please excuse our slight delay in providing the promised missing experimental results. We noticed a small mistake in our evaluation that led to the previously presented results of our answers only being evaluated with 3, namely the short-running seeds, instead of the full 5 seeds. In the following, we provide the updated tables. Our results remain qualitatively unchanged: highlighting the insufficiency of BNs for marginal inference tasks for our setups even more strongly. Similarly, we often outperform the applied causal normalizing flows in terms of L1/L2 errors, mainly due to strong prediction outliers and their general inability to perform exact marginal inference, c.f. Table 3.

This response provides all previously missing evaluation results. We hope this addresses any remaining concerns and are happy to answer any further questions.

Best,
the authors

---

|         | Method | 5/5         | 10/12       | 15/20       | 20/30       | 50/100      | 100/250     |
| ------- | ------ | ----------- | ----------- | ----------- | ----------- | ----------- | ----------- |
| **ID**  | cf-SPN | 0.98 ± 0.01 | 0.97 ± 0.02 | 0.99 ± 0.01 | N/A         | N/A         | N/A         |
|         | CNF    | 0.98 ± 0.01 | 0.99 ± 0.01 | 0.99 ± 0.01 | N/A         | N/A         | N/A         |
| **OOD** | cf-SPN | 1.00 ± 0.01 | 0.99 ± 0.00 | 0.99 ± 0.01 | 0.99 ± 0.00 | 1.00 ± 0.00 | 0.96 ± 0.01 |
|         | CNF    | 0.99 ± 0.01 | 0.99 ± 0.00 | 0.99 ± 0.01 | 0.99 ± 0.00 | 1.00 ± 0.00 | 1.00 ± 0.00 |

**Table: 2** **Element-Wise Accuracy Comparing cf-SPNs and CNFs.**

---

|         | Method | 5/5             | 10/12           | 15/20       | 20/30       | 50/100      | 100/250     |
| ------- | ------ | --------------- | --------------- | ----------- | ----------- | ----------- | ----------- |
| **ID**  | cf-SPN | 1.36 ± 0.52     | 1.52 ± 0.75     | 0.49 ± 0.56 | N/A         | N/A         | N/A         |
|         | CNF    | 10.68 ± 10.11   | 1.66 ± 0.34     | 0.70 ± 0.80 | N/A         | N/A         | N/A         |
| **OOD** | cf-SPN | 0.69 ± 0.90     | 1.00 ± 0.27     | 0.90 ± 0.37 | 0.69 ± 0.12 | 0.36 ± 0.09 | 3.47 ± 0.93 |
|         | CNF    | 167.03 ± 234.86 | 281.47 ± 543.49 | 4.32 ± 3.23 | 2.65 ± 2.08 | 1.30 ± 0.76 | 0.29 ± 0.09 |

**Table 3:** **L2-Errors of Marginal Probabilities for cf-SPNs and CNFs (scaled by 100).**

---

| Method | 5/5         | 10/12       | 15/20       | 20/30       | 50/100       | 100/250         |
| ------ | ----------- | ----------- | ----------- | ----------- | ------------ | --------------- |
| cf-SPN | 1.06 ± 0.10 | 2.02 ± 0.02 | 3.12 ± 0.04 | 3.98 ± 0.13 | 9.86 ± 0.20  | 20.05 ± 0.19    |
| CBN    | 1.16 ± 0.05 | 2.30 ± 0.05 | 3.62 ± 0.07 | 4.80 ± 0.12 | 15.32 ± 0.28 | 760.24 ± 673.32 |
| CNF    | 0.32 ± 0.03 | 0.50 ± 0.01 | 0.69 ± 0.01 | 0.84 ± 0.02 | 1.84 ± 0.03  | 3.60 ± 0.04     |

**Table 4:** **Average Inference Runtimes for CBNs and cf-SPNs (seconds).**

---

|         | Method | 5/5           | 10/12         | 15/20       | 20/30       | 50/100      | 100/250     |
| ------- | ------ | ------------- | ------------- | ----------- | ----------- | ----------- | ----------- |
| **ID**  | cf-SPN | 4.65 ± 1.53   | 4.33 ± 1.20   | 2.46 ± 1.32 | N/A         | N/A         | N/A         |
|         | CNF    | 8.23 ± 4.02   | 4.57 ± 0.38   | 3.03 ± 1.88 | N/A         | N/A         | N/A         |
| **OOD** | cf-SPN | 3.56 ± 2.43   | 3.89 ± 0.84   | 3.43 ± 1.07 | 2.80 ± 0.26 | 1.83 ± 0.28 | 5.65 ± 1.05 |
|         | CNF    | 21.35 ± 24.70 | 12.26 ± 12.05 | 5.22 ± 1.94 | 3.46 ± 0.48 | 1.93 ± 0.34 | 1.14 ± 0.04 |

**Table 9:** **L1-Errors of Marginal Probabilities for cf-SPNs and CNFs (scaled by 100).**

---

|         | Method | 5/5         | 10/12       | 15/20       | 20/30       | 50/100      | 100/250        |
| ------- | ------ | ----------- | ----------- | ----------- | ----------- | ----------- | -------------- |
| **ID**  | cf-SPN | 0.89 ± 0.04 | 0.81 ± 0.08 | 0.89 ± 0.14 | N/A         | N/A         | N/A            |
|         | CNF    | 0.89 ± 0.05 | 0.86 ± 0.05 | 0.90 ± 0.16 | N/A         | N/A         | N/A            |
| **OOD** | cf-SPN | 0.98 ± 0.04 | 0.88 ± 0.02 | 0.88 ± 0.06 | 0.84 ± 0.04 | 0.84 ± 0.07 | 0.02(*) ± 0.03 |
|         | CNF    | 0.95 ± 0.06 | 0.93 ± 0.03 | 0.89 ± 0.06 | 0.90 ± 0.04 | 0.91 ± 0.04 | 0.92 ± 0.05    |

**Table 10:** **Accuracy Comparing cf-SPNs and CNFs.**

---

### Decision · Action_Editor_Jpdg · 2025-08-01

**Recommendation:** Accept as is

**Audience:**

Yes

**Audience Explanation:**

The paper develops a probabilistic circuit approach for evaluating queries in structural causal models (SCMs). Specifically, the current work focuses on different rungs, including "counterfactual" queries (as defined in rung 3 of Pearl's causality ladder). Such queries aim to estimate causal quantities that can be used for "what would have happened questions" (which may be relevant in law, and certain medical reasoning applications). Note: counterfactual queries share similarities with counterfactual explanations in terms of the ladder, but the focus is on estimating a causal quantity vs. explanations for such a quantity.

The paper has novel components and opens interesting future research around probabilistic circuits in Causal ML. As such, it follows the mission of TMLR to promote such research, which can be still in a more emergent stage. The ladder is also reflected in some assumptions, which may be considered to be somewhat stronger (e.g., access to counterfactual data for the above task).

Minor: Table 1 -- I would move the authors' method to the bottom for better readability

**Claims And Evidence:**

Yes

**Claims Explanation:**

The reviewers did an excellent job in providing constructive and thoughtful feedback. Further, they also assess the strengths and weaknesses of the current submission, which I will summarize below. The authors updated their work with new experiments, and the reviewers are happy. I think the evidence is generally strong.